# Generalized dilaton gravity in 2d

Daniel Grumiller[1*], Romain Ruzziconi[1†] and Céline Zwikel[1,2‡]

**1** Institute for Theoretical Physics, TU Wien,
Wiedner Hauptstrasse 8-10/136 A-1040 Vienna, Austria
**2** Perimeter Institute for Theoretical Physics,
31 Caroline Street North, Waterloo, Ontario, Canada N2L 2Y5

★ grumil@hep.itp.tuwien.ac.at, † romain.ruzziconi@tuwien.ac.at,
‡ czwikel@perimeterinstitute.ca

## Abstract

Generalized dilaton gravity in 2d is the most general consistent deformation of the Jackiw–Teitelboim model that maintains local Lorentz invariance. The action is generically not power-counting renormalizable, thus going beyond the class of models typically studied. Nevertheless, all these models are exactly soluble. We focus on a subclass of dilaton scale invariant models. Within this subclass, we identify a 3-parameter family of models that describe black holes asymptoting to AdS$_2$ in the UV and to dS$_2$ in the IR. Since these models could be interesting for holography, we address thermodynamics and boundary issues, including boundary charges, asymptotic symmetries and holographic renormalization.



# 1 Introduction

An efficient way to define physical models is to write down the most general action compatible with the desired field content, global and gauge symmetries, and other physical requirements, such as power-counting renormalizability. In a second step one can then try to deform the model, maintaining the number of field- and gauge-degrees of freedom, but allowing the symmetries to be modified, thereby obtaining a larger class of models. This is called "consistent deformation", for a review see [1] and for selected earlier work see e.g. [2,3] as well as [4] and refs. therein. Eventually one ends up with a rigid class of models, meaning that they can only be deformed into each other, but not into any model outside this class.

Our work focuses on dilaton gravity in two dimensions (2d). Its purpose is two-fold: first, we highlight and review that the most general consistent deformation of the Jackiw–Teitelboim (JT) model to another 2d dilaton gravity model is *not* given by the commonly used bulk action [5,6] (see also [7,8] and refs. therein),[1]

$$I[g_{\mu\nu}, X] = -\frac{\kappa}{4\pi} \int d^2x \sqrt{-g}\left(XR - U(X)(\partial X)^2 - 2V(X)\right),\tag{1}$$

but rather by an action that is not power-counting renormalizable in general

$$I[g_{\mu\nu}, X] = -\frac{\kappa}{4\pi} \int d^2x \sqrt{-g}\left(XR - 2\mathcal{V}(X, -(\partial X)^2)\right).\tag{2}$$

---

[1] The dilaton is denoted by $X$ and the gravitational coupling constant by $\kappa = \frac{1}{4G}$, where $G$ is the two-dimensional Newton constant.

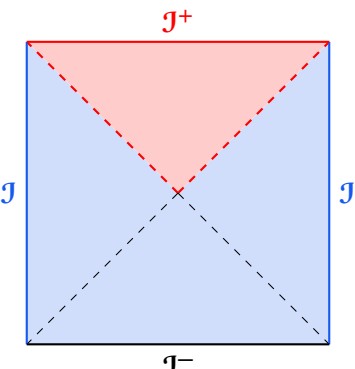

Figure 1: Penrose diagram of black hole (red) with $dS_2$ interior and $AdS_2$ asymptotics

Plausibly, there are two reasons why (2) is not widely known:[2] at first glance it seems meaningful to impose power-counting renormalizability as selection criterion for the model space. Note, however, that imposing power-counting renormalizability on a model space that describes theories without local physical degrees of freedom seems dubious (see e.g. the corresponding discussion of three-dimensional gravity [15]). Since dilaton gravity in 2d is a theory without local physical degrees of freedom we drop the requirement of power-counting renormalizability. The second reason is more mundane, but still relevant: so far there appear to be no interesting models (2) that do not reduce to the simpler class of models (1). Thus, our second goal is to provide such models. We focus particularly on a subclass of models that exhibit an extra symmetry at the level of equations of motion (EOM), namely dilaton scale invariance, meaning that under constant rescalings

$$X \to \lambda X, \qquad \lambda \in \mathbb{R}^+, \tag{3}$$

a solution to the classical EOM is mapped to another solution. Within the more restrictive class of models (1), JT gravity [16, 17] and the Witten black hole [18–20] are dilaton scale invariant.[3] As we shall see, within the model space defined by (2) there is an infinite class of such models, labelled by one free function of one variable.

Within this infinite class, we identify a 3-parameter family of models, the solutions of which asymptote to $AdS_2$ in the UV (from a dual field theory perspective) and to $dS_2$ in the IR, without the need for matter degrees of freedom, domain walls or other auxiliary constructions. Moreover, the solutions correspond to black holes, see Fig. 1 for a typical Penrose diagram.

Since these models may have useful applications in holography, the remainder of the paper is devoted to their detailed study, including boundary issues, asymptotic symmetries and holographic renormalization. We highlight here the most intriguing details of this analysis:

- All solutions are asymptotically $AdS_2$ (or asymptotically flat) for large dilaton and approach $dS_2$ for small dilaton. Almost all solutions have a horizon in between.

- There are simple expressions in terms of elementary functions for the Casimir of all these models, which on-shell corresponds to the mass of the (black hole) solution.

- The models are labelled by two essential parameters only, one of which is a discriminant. The (square root of the) discriminant has a geometric interpretation as Lifshitz exponent

---

[2]The action (2) has appeared, e.g., in [7, 9–12] and is also known as "kinetic gravity braiding" [13] or "Horndeski theory" [14]. As opposed to the action (1), so far few explicit examples were studied and did not involve holographic considerations.

[3]Since in a string theory context usually $\phi = -\frac{1}{2} \ln X$ is used as dilaton this property is also known as dilaton shift invariance, $\phi \to \phi - \frac{1}{2} \ln \lambda$ [10].

$z$, appearing in an asymptotic Killing vector associated with anisotropic scale invariance. Thus, the bulk asymptotic symmetries suggest Lifshitz-type of scaling behavior, which is corroborated by the Lifshitz-type scaling of entropy with mass. This is remarkable, since from a boundary perspective there is only one dimension, so there is nothing obvious with respect to which the boundary direction could be anisotropic.

- The value of the Ricci scalar at the horizon can have either sign, but even when positive, the near horizon region is never approximately dS$_2$ since the Ricci scalar varies too quickly. In a particular scaling limit we find a model that is locally AdS$_2$ almost everywhere outside the horizon, locally dS$_2$ almost everywhere inside the horizon and has a steep gradient of the Ricci scalar in a tiny region around the black hole horizon, somewhat reminiscent of the black hole droplet model in the large $D$ expansion where also large gradients appear very close to the horizon [21,22]. This limiting model could be tailor made for near horizon studies.

This paper is organized as follows. In section 2 we review the most general consistent deformation of JT gravity, which is a Poisson $\sigma$-model with 3-dimensional target space; in order for such a model to have a gravity interpretation we impose the requirement that local Lorentz invariance is part of the gauge symmetries, which then reduces the model space to (2) in the metric formulation. In section 3 we provide a detailed analysis of the phase space and relate the first and second order formulations. From section 4 onward we focus on dilaton scale invariant models, and more specifically on a simple but physically intriguing 3-parameter family therein. In section 5 we present a detailed discussion of the asymptotic and curvature behavior, as well as holographic renormalization and thermodynamical properties. A consistent set of boundary conditions for dilaton scale invariant models is discussed in section 6. In section 7 we conclude.

## 2 Generalized 2d dilaton gravity theory

In this section, we review how to construct the most general consistent deformation of JT gravity. In section 2.1 we recall the first order formulation of 2d dilaton gravity as a specific type of Poisson $\sigma$-model with extra structure. In section 2.2, after reviewing what precisely is a consistent deformation, we re-derive the most general consistent deformation of JT gravity. In section 2.4 we translate the results into the second order formulation.

### 2.1 First order formulation as nonlinear gauge theory

Dilaton gravity in 2d has a gauge theoretic formulation as Poisson $\sigma$-model [23,24], reminiscent of the Chern–Simons formulation of Einstein gravity in 3d [25,26]. Its bulk action,

$$I_{\text{PSM}}[A_I, X^I] = \frac{\kappa}{2\pi} \int \left( X^I \, \mathrm{d}A_I + \frac{1}{2} P^{IJ}(X^K) A_I \wedge A_J \right),$$

(4)

depends functionally on a set of target space coordinates $X^I$ that span a Poisson manifold with Poisson tensor $P^{IJ} = -P^{JI}$, subject to the non-linear Jacobi identity[4]

$$P^{IL} \, \partial_L P^{JK} + P^{JL} \, \partial_L P^{KI} + P^{KL} \, \partial_L P^{IJ} = 0 \,.$$

(5)

---

[4]The Jacobi identity (5) allows to interpret the target space as being non-commutative, in the sense that the Jacobi identity for the Schouten–Nijenhuis bracket $\{X^I, X^J\} = P^{IJ}$ is identical to (5). This target space non-commutativity features prominently in the Kontsevich $\star$-product [27], but will not play a decisive role in the present work.

The gauge field 1-forms $A_I$ and the target space coordinates transform in general non-linearly,

$$\delta_\lambda X^I = \lambda_J P^{JI}, \qquad \delta_\lambda A_I = d\lambda_I + \partial_I P^{JK} A_J \lambda_K, \qquad (6)$$

under gauge transformations that preserve the action (4). Whenever the Poisson tensor is linear in the target space coordinates the gauge symmetries (6) are of Lie-algebra type, with $\partial_I P^{JK}$ being the structure constants. Finally, the coupling constant $\kappa$ plays a role similar to the Chern–Simons level in 3d; for theories with a gravity interpretation $\kappa$ is proportional to the inverse 2d Newton constant.

The EOM derived from the action (4)

$$dX^I + P^{IJ} A_J = 0, \qquad dA_I + \frac{1}{2}\left(\partial_I P^{JK}\right) A_J \wedge A_K = 0, \qquad (7)$$

are first order non-linear PDEs. Since the action (4) does not depend on any background metric it is an example of a topological quantum field theory of Schwarz type, see e.g. [28]. Moreover, Poisson $\sigma$-models have no local physical degrees of freedom, so their physical phase spaces can be interpreted holographically as boundary phase spaces. All these properties are shared with 3d Chern–Simons theories.

JT gravity is a special case of a Poisson $\sigma$-model where the target space manifold is three-dimensional and the Poisson tensor is linear in the target space coordinates. As a consequence of this linearity we are back to the realm of non-abelian gauge theories. Indeed, JT gravity can be understood as a non-abelian BF theory with gauge group SL(2, $\mathbb{R}$) [29,30] (or some restriction thereof, see [31]).

The gauge field content of JT gravity comprises zweibein $e_a$ and (dualized) spin-connection $\omega$, $A_I = (\omega, e_a)$. The target space coordinates, $X^I = (X, X^a)$ are physically interpreted as dilaton field ($X$) and Lagrange-multipliers for the torsion constraints ($X^a$). In terms of these variables the JT gravity action,

$$I_{\text{JT}}[\omega, e_a, X, X^a] = \frac{\kappa}{2\pi} \int \left( X \, d\omega + X^a \left( de_a + \epsilon_a{}^b \, \omega \wedge e_b \right) + \frac{\Lambda}{2} \epsilon^{ab} e_a \wedge e_b X \right), \qquad (8)$$

has a natural geometric interpretation: variation with respect to $X^a$ establishes vanishing torsion on-shell, and variation with respect to the dilaton $X$ yields a curvature 2-form proportional to the volume 2-form, and hence constant curvature solutions. Comparison between (8) and (4) shows that for JT gravity the Poisson tensor is of the form

$$P_{\text{JT}}^{IJ}(X, X^+, X^-) = \begin{pmatrix} 0 & -X^+ & X^- \\ X^+ & 0 & \Lambda X \\ -X^- & -\Lambda X & 0 \end{pmatrix}, \qquad (9)$$

where we used light-cone gauge for the Minkowski metric, $\eta_{+-} = 1$, $\eta_{\pm\pm} = 0$, and for the indices $a, b$. Our sign convention is $\epsilon^\pm{}_\pm = \pm 1$. Note that $X^a X_a = 2X^+ X^-$.

In JT gravity the three gauge symmetries generated by $\lambda_I = (\lambda_\omega, \lambda_a)$ have a simple geometric interpretation: $(\lambda_\omega, 0)$ generates local Lorentz transformations

$$\delta_{\lambda_\omega} \omega = d\lambda_\omega, \qquad \delta_{\lambda_\omega} e^\pm = \mp\lambda_\omega e^\pm, \qquad \delta_{\lambda_\omega} X = 0, \qquad \delta_{\lambda_\omega} X^\pm = \mp\lambda_\omega X^\pm, \qquad (10)$$

while $\lambda_a$ (together with a compensating Lorentz transformation) generates gauge transformations that on-shell correspond to 2d diffeomorphisms generated by some vector field $\xi^\mu$, such that $\lambda_I = A_{I\mu} \xi^\mu$. This identification yields

$$\begin{aligned} \delta_\xi e^a_\mu &\approx \xi^\nu \partial_\nu e^a_\mu + e^a_\nu \partial_\mu \xi^\nu, & \delta_\xi X^a &\approx \xi^\nu \partial_\nu X^a, \\ \delta_\xi \omega_\mu &\approx \xi^\nu \partial_\nu \omega_\mu + \omega_\nu \partial_\mu \xi^\nu, & \delta_\xi X &\approx \xi^\nu \partial_\nu X, \end{aligned} \qquad (11)$$

where the similarity sign $\approx$ denotes on-shell equivalence, using the condition of vanishing torsion (constant curvature) in the first (second) line. This is again in analogy to 3d gravity in the Chern–Simons formulation, see e.g. [32].

Since we are interested in the most general consistent deformation of JT gravity we can phrase what we want to achieve as follows: we search for the most general consistent deformation of 2d BF theory that still allows for a gravity interpretation.

## 2.2 Most general consistent deformation of JT gravity

Consistent deformations (in the sense of [2]) allow to deform the gauge symmetries, but do not change the number of field- or gauge-degrees of freedom. Thus, consistent deformations maintain the number of local physical degrees of freedom. Since JT gravity has no local physical degree of freedom, its most general consistent deformation also has this property. JT gravity is a non-abelian BF theory (which in turn is a consistent deformation of abelian BF theory). We are thus interested in the most general consistent deformation of 2d BF theory.

Izawa showed that the most general consistent deformation of 2d BF theory is a Poisson $\sigma$-model with the same dimension of the target space [33]. The gist of the proof is as follows. Any deformed Lagrangian $\tilde{\mathcal{L}}$ must obey the consistency relation ($s$ and d are BRST and de Rham-differential, respectively)

$$s\tilde{\mathcal{L}} + \mathrm{d}\tilde{a}_1 = 0 \tag{12}$$

together with the descent equations $s\tilde{a}_1 + \mathrm{d}\tilde{a}_0 = s\tilde{a}_0 = 0$ (they all follow from the master equation, see [2]). Starting from 2d abelian BF theory, Izawa constructed the BRST differential $s$, found the most general solutions for the bottom of the descent ladder, the ghost-number 2 term $\tilde{\tilde{a}}_0$, as well as the higher steps in the ladder, $\tilde{a}_0$ and $\tilde{\mathcal{L}} = \frac{1}{2} P^{IJ}(X^K) A_I \wedge A_J$ (up to terms containing antifields), where consistency of the deformation demands the Jacobi identities (5). Thus, the class of Poisson $\sigma$-models (4) with fixed target space dimension is rigid, i.e., no consistent deformation can move us out of this class.

This is almost the solution we want. However, as stressed before we need a bit more structure to interpret a Poisson $\sigma$-model as gravity theory. In particular, we demand local Lorentz invariance. This means that the most general 2d dilaton gravity model that emerges as consistent deformation from JT gravity must have a Poisson tensor of the form

$$P^{IJ} = \begin{pmatrix} 0 & -X^+ & X^- \\ X^+ & 0 & \mathcal{V}(X, X^+, X^-) \\ -X^- & -\mathcal{V}(X, X^+, X^-) & 0 \end{pmatrix}, \tag{13}$$

where $\mathcal{V}$ is an arbitrary function of all three target space coordinates, subject to constraints imposed by the Jacobi identities (5). Before solving these constraints let us stress that the first row and column of the Poisson tensor must be as given in (13) in order for (10) to hold. Moreover, an immediate consequence of the anti-symmetry, $P^{IJ} = -P^{JI}$, is the existence of a vanishing eigenvalue of the $3 \times 3$-matrix (13). Associated with this vanishing eigenvalue is the existence of a Casimir function $C(X, X^+, X^-)$ that is absolutely conserved on-shell, $\mathrm{d}C = 0$ (see e.g. [9]). Its physical interpretation in the context of 2d dilaton gravity is as mass of the state, e.g., the black hole mass [5, 34, 35]. We shall construct the Casimir function explicitly in examples below.

The last two terms in the Jacobi identities (5)

$$P^{X+}\partial_+\mathcal{V} + P^{X-}\partial_-\mathcal{V} + \mathcal{V}\partial_- P^{-X} - \mathcal{V}\partial_+ P^{X+} = 0 \tag{14}$$

always cancel, but the first two terms only cancel provided the potential only depends on two independent arguments, $\mathcal{V}(X, X^+, X^-) = \mathcal{V}(X, 2X^+ X^-)$. Rephrasing this result in a frame-independent way, the most general function $\mathcal{V}(X, X^a)$ compatible with the Jacobi identities

can only depend on Lorentz invariant combinations

$$\mathcal{V}(X, X^a) = \mathcal{V}(X, X^a X_a). \tag{15}$$

Like for JT gravity, Latin indices are raised and lowered with the Minkowski metric $\eta_{ab}$.

In conclusion, the most general 2d dilaton gravity theory that has the same number of gauge- and field-degrees of freedom as JT gravity is given by the first order action

$$I_{\text{gen}}[e_a, \omega, X, X^a] = \frac{\kappa}{2\pi} \int \left( X \, d\omega + X^a \left( de_a + \epsilon_a{}^b \, \omega \wedge e_b \right) + \frac{1}{2} \epsilon^{ab} e_a \wedge e_b \, \mathcal{V}(X, X^c X_c) \right). \tag{16}$$

The EOM are

$$dX + X^a \epsilon_a{}^b e_b = 0, \tag{17a}$$

$$dX^a - X_b \epsilon^{ba} \omega + \epsilon^{ab} e_b \mathcal{V} = 0, \tag{17b}$$

$$d\omega + \frac{1}{2} \epsilon^{ab} e_a \wedge e_b \, \partial_X \mathcal{V} = 0, \tag{17c}$$

$$de_a + \epsilon_a{}^b \, \omega \wedge e_b + \frac{1}{2} \epsilon^{cb} e_c \wedge e_b \, \partial_{X^a} \mathcal{V} = 0. \tag{17d}$$

We shall refer to this model as "generalized dilaton gravity".

## 2.3 Solving the equations of motion

The solutions of the EOM (17) fall into two classes, linear and constant dilaton vacua. We start with the former.

### 2.3.1 Linear dilaton vacua

We now derive all solutions in the linear dilaton regime for the generalized dilaton gravity theory in the first order formulation (16). To do so, it is convenient to write the EOM (17) in the light-cone gauge introduced above. The potential (15) can be written as a function of $X$ and

$$\tilde{X} := \frac{X^a X_a}{2X^2} = \frac{X^+ X^-}{X^2}. \tag{18}$$

The EOM

$$dX + X^- e^+ - X^+ e^- = 0, \tag{19a}$$

$$(d \pm \omega) X^\pm \pm e^\pm \mathcal{V} = 0, \tag{19b}$$

$$d\omega + \epsilon \frac{\partial \mathcal{V}}{\partial X} = 0, \tag{19c}$$

$$(d \pm \omega) e^\pm + \epsilon \frac{\partial \mathcal{V}}{\partial X^\mp} = 0, \tag{19d}$$

contain the volume form $\epsilon = \frac{1}{2} \epsilon^{ab} e_a \wedge e_b = e^- \wedge e^+$.

Taking a linear combination of the two equations in (19b), multiplied, respectively, by $X^-$ and $X^+$, and then inserting (19a), obtains $d(X^+ X^-) - \mathcal{V} dX = 0$ or, in terms of $\tilde{X}$ defined in (18),

$$d\tilde{X} + \left( 2\tilde{X} - \frac{\mathcal{V}}{X} \right) \frac{dX}{X} = 0. \tag{20}$$

The relation (20) implies Casimir conservation $dC = 0$ and can be integrated to yield the Casimir function $C(X, \tilde{X})$. However, we postpone doing so and focus on solving the other EOM first.

We assume now $X^+ \neq 0$.[5] The equation (19b) implies

$$\omega = -\frac{\mathrm{d}X^+}{X^+} - Z\mathcal{V}, \tag{21}$$

where $Z = e^+/X^+$. Similarly, the equation (19a) yields

$$e^- = \frac{\mathrm{d}X}{X^+} + X^- Z, \tag{22}$$

and the volume element $\epsilon = -Z \wedge \mathrm{d}X$. Combining the upper signs (19d) and (19b) obtains

$$\mathrm{d}Z = (Z \wedge \mathrm{d}X)\frac{1}{X^2}\frac{\partial \mathcal{V}}{\partial \tilde{X}}. \tag{23}$$

Inserting $Z = \mathrm{d}v\, e^{Q(X)}$ into this equation yields

$$\frac{\mathrm{d}Q}{\mathrm{d}X} = -\frac{1}{X^2}\frac{\partial \mathcal{V}}{\partial \tilde{X}}, \tag{24}$$

which can be formally integrated as $Q(X) = -\int^X \frac{1}{X^2}\partial_{\tilde{X}}\mathcal{V}$ by virtue of (20). The line element reads as

$$\mathrm{d}s^2 = 2e^+e^- = 2e^Q\, \mathrm{d}v\, \mathrm{d}X + 2\tilde{X}e^{2Q}X^2\, \mathrm{d}v^2, \tag{25}$$

where $\tilde{X}$ is determined by (20) and $Q(X)$ by (24). The solution space is parametrized by two constants of integration. The one coming from the integration of (20) is non-trivial and related to the Casimir of the theory, while the one coming from (24) is trivial and can be fixed by a choice of units. Note that $Q(X)$ can depend on the Casimir in generalized dilaton gravity models, in contrast to the commonly studied models (1). In appendix A, we specialize the algorithm above to the power-counting renormalizable models (1), where we explicitly perform the integrals related to Eqs. (20) and (24), thereby recovering well-known results.

As may have been anticipated (see e.g. [38]), all linear dilaton solutions for all generalized dilaton gravity models obey a generalized Birkhoff theorem, in the sense that all solutions (25) exhibit a Killing vector $\partial_v$.

### 2.3.2 Constant dilaton vacua

In addition to the linear dilaton solutions discussed above, there can be a constant dilaton sector, provided the non-differential equations

$$\mathcal{V}(X, X^a X_a) = 0 = X^a \tag{26}$$

have solutions. All these solutions have constant dilaton, constant curvature and are locally maximally symmetric. Since these solutions do not differ in any essential way from the well-known constant dilaton vacua of ordinary 2d dilaton gravity (see e.g. [39]), we shall not discuss them in our work.

---

[5]This choice implies Eddington–Finkelstein gauge for the metric. Since the EOM are symmetric with respect to $X^- \leftrightarrow X^+$, we could choose $X^- \neq 0$ instead in case $X^+ = 0$, which corresponds to changing from in- to outgoing Eddington–Finkelstein gauge. If both vanish in an open region, $X^{\pm} = 0$, we obtain a constant dilaton solution instead. If both vanish on an isolated point then our coordinate system breaks down, which happens at bifurcation points of Killing horizons. To provide a local chart containing the bifurcation point one can use a Kruskal-type of coordinate system, required only in the near horizon approximation. See [36,37] for more details.

## 2.4 Translation to second order formulation

Following the standard procedure of integrating out the auxiliary fields in (16) (see e.g. [7]), one can rewrite the generalized 2d dilaton gravity theory in the second order metric formalism. We briefly summarize these steps.

In terms of the zweibein and the spin-connection, the curvature and the torsion 2-forms read as $\mathcal{R}_{ab} = 2\,\mathrm{d}\omega\,\epsilon_{ab}$ and $\mathcal{T}_a = \mathrm{d}e_a + \epsilon_a{}^b\omega \wedge e_b$, respectively. We denote the torsion-free part of the spin-connection by $\tilde{\omega}$, i.e., $T_a(\tilde{\omega}) = 0$. The spin-connection can be decomposed as

$$\omega = \tilde{\omega} - e^a \mathcal{T}_{a\mu\nu}\epsilon^{\mu\nu}, \tag{27}$$

where $\tilde{\omega} = e^a(\partial_\mu e_{a\nu})\epsilon^{\mu\nu}$ or equivalently $\tilde{\omega}_\mu\epsilon^a{}_b = e^a_\nu\nabla_\mu e^\nu_b$, and we also define $R_{ab\mu\nu} = (\partial_\mu\tilde{\omega}_\nu - \partial_\nu\tilde{\omega}_\mu)\epsilon_{ab}$. The EOM (17d) and (17a) yield

$$\mathcal{T}_a = -\frac{1}{2}\epsilon^{cb}e_c \wedge e_b \partial_{X^a}\mathcal{V}, \qquad X^a = -e^a_\nu\epsilon^{\mu\nu}\partial_\mu X, \tag{28}$$

and hence $\tilde{X} = -(\partial X)^2/(2X^2)$. Injecting these expressions into the first-order action (16) to eliminate the auxiliary fields $\omega$ and $X^a$ establishes the second order action

$$I_{\text{gen}}[g_{\mu\nu}, X] = -\frac{\kappa}{4\pi}\int \mathrm{d}^2 x\,\sqrt{-g}\Big(RX - 2\mathcal{V}\big(X, -(\partial X)^2\big)\Big), \tag{29}$$

where $g_{\mu\nu} = e^a_\mu\eta_{ab}e^\nu_b$ is the spacetime metric and $R = R_{ab\mu\nu}e^{a\mu}e^{b\nu} = (\partial_\mu\tilde{\omega}_\nu - \partial_\nu\tilde{\omega}_\mu)\epsilon^{\mu\nu}$ is the Ricci scalar. Note that we have thrown away a boundary term

$$I^{\text{first order}}_{\text{gen}} = I^{\text{second order}}_{\text{gen}} + \frac{\kappa}{\pi}\int \mathrm{d}^2 x\,\partial_\mu\big(X\sqrt{-g}\nabla^\mu X\partial_\nabla\mathcal{V}\big). \tag{30}$$

The EOM for the theory (29)

$$\nabla_\mu\nabla_\nu X - g_{\mu\nu}\nabla^2 X - 2(\partial_\mu X)(\partial_\nu X)(\partial_\nabla\mathcal{V}) - \mathcal{V}g_{\mu\nu} = 0, \tag{31a}$$

$$R + 8(\nabla^\mu\nabla^\nu X)(\partial_\mu X)(\partial_\nu X)\partial^2_\nabla\mathcal{V} - 4(\nabla^2 X)\partial_\nabla\mathcal{V} - 4(\partial X)^2\partial_\nabla\partial_X\mathcal{V} - 2\partial_X\mathcal{V} = 0, \tag{31b}$$

are non-linear second order PDEs. We used the notation $\partial_\nabla := -\frac{\partial}{\partial((\partial X)^2)}$ and $\partial_X := \frac{\partial}{\partial X}$.

# 3 Phase space for generalized dilaton gravity

In this section, we analyze the phase space of generalized dilaton gravity in 2d and provide the general expressions for the charges using the standard methods of covariant phase space formalism [40–44]. We present the results in both first and second order formulations and then relate them.

## 3.1 First order formalism

We denote the dynamical fields of the theory summarily by $\phi = (e_a, \omega, X, X^a)$. Keeping the boundary terms in the variation of the action (16) when deriving the EOM obtains the canonical presymplectic potential

$$\delta I_{\text{gen}}[\phi] = (\text{EOM})\,\delta\phi + \int_{\partial M}\Theta_{1,\text{gen}}[\phi, \delta\phi], \qquad \Theta_{1,\text{gen}}[\phi, \delta\phi] = \frac{\kappa}{2\pi}(X_a\,\delta e^a + X\,\delta\omega). \tag{32}$$

The index 1 stands for first order. The presymplectic current is obtained by taking a variation of the presymplectic potential

$$\omega_{1,\mathrm{gen}}[\phi,\delta\phi,\delta'\phi] = \frac{\kappa}{2\pi}\left(\delta X_a\,\delta'e^a + \delta X\,\delta'\omega\right) - (\delta \leftrightarrow \delta'). \tag{33}$$

The symmetries of the action (16) are given by the diffeomorphisms parametrized by vector fields $\xi^\mu$ and the so(1,1) local Lorentz transformations parametrized by $\lambda_\omega$. The infinitesimal variations of the fields are given in (10) and (11).

The co-dimension 2-form of the theory associated with these symmetries, which is a 0-form in two dimensions, can be deduced by contracting the presymplectic current (33) with an infinitesimal symmetry. We obtain

$$k_{\xi,\lambda_\omega}[\phi,\delta\phi] = -\frac{\kappa}{2\pi}\left(e_\rho^a\,\xi^\rho\,\delta X_a + (\omega_\rho\,\xi^\rho + \lambda_\omega)\,\delta X\right). \tag{34}$$

Since the theory (16) is a first order covariantized Hamiltonian theory in the sense of [45], there is no ambiguity in the definition of the charges at this level, and the Barnich–Brandt [42, 43] and Iyer–Wald [40, 41] procedures coincide with each other.

## 3.2 Second order formalism

In the second order formulation (29), the dynamical fields are given by $\phi = (g_{\mu\nu}, X)$. The boundary terms obtained by integrating by parts in the variation of the action (29) to obtain (31) define the canonical presymplectic potential of the theory,

$$\Theta_{2,\mathrm{gen}}^\mu[\phi,\delta\phi] = -X\Theta_{\mathrm{EH}}^\mu[g,\delta g] - \frac{\kappa}{4\pi}\sqrt{-g}\left(\nabla^\mu X(\delta g)_\nu^\nu - (\delta g)_\nu^\mu\nabla^\nu X + 4(\partial_\nabla\mathcal{V})\delta X(\nabla^\mu X)\right), \tag{35}$$

with the Einstein–Hilbert presymplectic potential

$$\Theta_{\mathrm{EH}}^\mu[g,\delta g] = \frac{\kappa}{4\pi}\sqrt{-g}\left(\nabla_\nu(\delta g)^{\mu\nu} - \nabla^\mu(\delta g)_\nu^\nu\right). \tag{36}$$

The index 2 in (35) stands for second order. One can then derive the presymplectic current as $\omega_{2,\mathrm{gen}}^\mu[\phi,\delta\phi,\delta'\phi] = \delta\Theta_{2,\mathrm{gen}}^\mu[\phi,\delta'\phi] - (\delta\leftrightarrow\delta')$. The symmetries of the generalized dilaton gravity theory (29) are given by the diffeomorphisms which act through Lie derivatives on the fields, i.e., $\delta_\xi g_{\mu\nu} = \mathcal{L}_\xi g_{\mu\nu}$ and $\delta_\xi X = \mathcal{L}_\xi X$. The Barnich–Brandt co-dimension 2-form is given by

$$k_{\mathrm{BB},\xi}^{\mu\rho}[\phi,\delta\phi] = \frac{\kappa}{2\pi}\sqrt{-g}\left(2(\nabla^{[\mu}\delta X)\xi^{\rho]} - 4\delta X(\partial_\nabla\mathcal{V})\xi^{[\rho}(\nabla^{\mu]}X)\right.$$
$$\left. - \delta X\nabla^{[\mu}\xi^{\rho]} + \xi^{[\mu}\delta g_\alpha^{\rho]}\nabla^\alpha X\right). \tag{37}$$

The relation between Barnich–Brandt [43, 44] and Iyer–Wald [40, 41] procedures is controlled by the object

$$E^{\mu\nu}[\phi;\delta\phi,\delta'\phi] = \frac{\kappa}{4\pi}\sqrt{-g}\,X\,\delta g_\sigma^\mu\,\delta'g^{\nu\sigma} - (\delta\leftrightarrow\delta'), \tag{38}$$

so that

$$k_{\mathrm{BB},\xi}^{\mu\nu}[\phi;\delta\phi] = k_\xi^{\mu\nu}[\phi;\delta\phi] - E^{\mu\nu}[\phi;\delta_\xi\phi,\delta\phi], \tag{39}$$

where $k_\xi^{\mu\nu}[\phi;\delta\phi]$ stands for the Iyer–Wald co-dimension 2-form. On-shell, we have the conservation law

$$\partial_\nu k_\xi^{\mu\nu}[\phi;\delta\phi] = -\omega^\mu[\phi;\delta_\xi\phi,\delta\phi]. \tag{40}$$

The results presented in this section consistently reduce to those established in [46] when restricting to the class of models (1).

### 3.3   Relation between first and second order symplectic potentials

The presymplectic potential (35) obtained by varying the action (29) in the second order formulation is related to the one of the first order formulation obtained in (32) through

$$\Theta^{\mu}_{2,\text{gen}}[\phi,\delta\phi] + \delta\left[\frac{\kappa}{\pi}\sqrt{-g}X\nabla^{\mu}X\partial_{\nabla}\mathcal{V}\right] = \Theta^{\mu}_{1,\text{gen}}[\phi,\delta\phi] + \partial_{\nu}\left[\frac{\kappa}{2\pi}|e|Xe^{[\mu}_{a}\delta e^{\nu]a}\right], \tag{41}$$

when setting the auxiliary fields on-shell in the first order formulation and writing $|e| = |\det(e^{a}_{\mu})| = \sqrt{-g}$. The $\delta$-exact term in the left-hand side comes from the relative boundary term between first and second order actions that we have thrown away in (30). This term plays no role in the charges. However, the relative corner term $\partial_{\nu}Y^{\mu\nu}$, with

$$Y^{\mu\nu}[\phi,\delta\phi] = \frac{\kappa}{2\pi}|e|Xe^{[\mu}_{a}\delta e^{\nu]a}, \tag{42}$$

may generically bring a contribution to the charges for a given set of boundary conditions, depending on whether we work in the first or the second order formulation. We will discuss an explicit example in appendix B. Note that (42) is of the same form as the corner term found in higher-dimensional Einstein gravity when relating first and second order formulations [47,48], up to the presence of the dilaton.

The corner (42) can be rewritten as

$$Y^{\mu\nu}[\phi,\delta\phi] = \frac{\kappa}{4\pi}\epsilon_{ab}\epsilon^{\mu\nu}Xe^{a\rho}\delta e^{b}_{\rho}, \tag{43}$$

using $e^{\mu}_{a} = -\epsilon^{\mu\nu}\epsilon_{ab}e^{b}_{\nu}$.

## 4   Dilaton scale invariant models

In this section we focus on specific dilaton scale invariant models, which we define in section 4.1. In section 4.2 we solve the classical EOM explicitly.

### 4.1   Potential for dilaton scale invariant models

Among the generalized dilaton gravity theories (2) there is a particular subclass for which the potential takes the form

$$\mathcal{V}(X,X^{a}X_{a}) = -X\mathcal{U}(\tilde{X}), \quad \text{with } \tilde{X} := \frac{X^{+}X^{-}}{X^{2}}. \tag{44}$$

All models in this subclass are scale invariant in the following sense. Rescaling the dilaton as ($\lambda \in \mathbb{R}$)

$$X^{\pm} \to \lambda X^{\pm}, \qquad X \to \lambda X, \tag{45}$$

the action (2) transforms multiplicatively,

$$I_{\text{gen}} \to \lambda I_{\text{gen}}. \tag{46}$$

This rescaling is therefore a symmetry of the EOM, but not of the Lagrangian. (See [49] for a discussion of such symmetries.)

To investigate the properties of scale invariant models, we take the simplest expression for the potential that allows to include new models not described by the subclass (1). More specifically, we consider quadratic polynomials for the potential $\mathcal{U}$ in (44),

$$\mathcal{U}(\tilde{X}) = a_{0} + a_{1}\tilde{X} + a_{2}\tilde{X}^{2}, \tag{47}$$

where $a_0, a_1, a_2$ are real constants.

The case $a_0 \neq 0$, $a_1 = 0$, $a_2 = 0$ corresponds to JT and $a_0 \neq 0$, $a_1 = -2$, $a_2 = 0$ to the Witten black hole. More generally, all cases with $a_2 = 0$ are included in (1), but not $a_2 \neq 0$. Therefore, the case $a_2 \neq 0$ is a non-trivial generalization of (1). From now on we assume $a_2 \neq 0$.

In summary, the remainder of this work is devoted to the study of generalized dilaton gravity models (2) with the 3-parameter potential

$$\mathcal{V} = -a_0 X + a_1 \frac{(\partial X)^2}{2X} - a_2 \frac{(\partial X)^4}{4X^3}, \qquad a_2 \neq 0. \tag{48}$$

## 4.2 Solutions of dilaton scale invariant models

In this section, we specialize the generic solution (25) derived in section 2.3 to the potential (44) and then specify it for the particular case (47). The equation (20) for $\tilde{X}$ can be integrated as

$$\ln \frac{X}{C} = -\int^{\tilde{X}} \frac{\mathrm{d}\tilde{Y}}{\mathcal{U}(\tilde{Y}) + 2\tilde{Y}} \stackrel{(47)}{=} -\int^{\tilde{X}} \frac{\mathrm{d}\tilde{Y}}{a_0 + (a_1 + 2)\tilde{Y} + a_2 \tilde{Y}^2}, \tag{49}$$

where $C$ is an integration constant corresponding to the on-shell value of the Casimir. The equation (24) for $Q$ becomes

$$\frac{\mathrm{d}Q}{\mathrm{d}X} = \frac{1}{X} \frac{\partial \mathcal{U}}{\partial \tilde{X}} \stackrel{(47)}{=} \frac{1}{X} \left( a_1 + 2a_2 \tilde{X} \right). \tag{50}$$

We start by solving the equation for $\tilde{X}$ (49), depending on the sign of the discriminant

$$\Delta := (a_1 + 2)^2 - 4a_0 a_2. \tag{51}$$

### 4.2.1 Positive discriminant

For $\Delta > 0$, equation (49) leads to

$$\tilde{X} = -\frac{a_1 + 2}{2a_2} + \frac{\sqrt{\Delta}}{2a_2} \tanh\left[ \frac{\sqrt{\Delta}}{2} \ln \frac{X}{C} \right], \tag{52}$$

where $C$ will be related to the mass below. Furthermore, equation (50) leads to

$$X \frac{\mathrm{d}Q}{\mathrm{d}X} = \sqrt{\Delta} \tanh\left[ \frac{\sqrt{\Delta}}{2} \ln \frac{X}{C} \right] - 2, \tag{53}$$

which can be integrated to give

$$e^Q = \frac{b^2}{X^2} \cosh^2\left[ \frac{\sqrt{\Delta}}{2} \ln \frac{X}{C} \right], \tag{54}$$

where $b$ is a constant. Inserting these results into the metric (25), we find explicitly

$$g_{vX} = \frac{b^2 C^{\sqrt{\Delta}}}{4X^{\sqrt{\Delta}+2}} \left( \frac{X^{\sqrt{\Delta}}}{C^{\sqrt{\Delta}}} + 1 \right)^2, \tag{55a}$$

$$g_{vv} = -\frac{b^4 C^{2\sqrt{\Delta}}}{16a_2 X^{2\sqrt{\Delta}+2}} \left( \frac{X^{\sqrt{\Delta}}}{C^{\sqrt{\Delta}}} + 1 \right)^3 \left( \left( a_1 + 2 - \sqrt{\Delta} \right) \frac{X^{\sqrt{\Delta}}}{C^{\sqrt{\Delta}}} + a_1 + 2 + \sqrt{\Delta} \right). \tag{55b}$$

The case of positive discriminant will be studied in more detail in section 5.

### 4.2.2 Vanishing discriminant

For $\Delta = 0$, one can set $a_0 = (a_1 + 2)^2/(4a_2)$. Solving (49), one has

$$\tilde{X} = \frac{1}{a_2 \ln \frac{X}{C}} - \frac{a_1 + 2}{2a_2}. \tag{56}$$

Note that this solution does not correspond to the limit $\Delta \to 0$ of the case $\Delta > 0$ presented above. Indeed,

$$\lim_{\Delta \to 0} \tilde{X}_{\Delta > 0} = -\frac{a_1 + 2}{2a_2} \tag{57}$$

is a constant. Therefore, one has to treat this sub-case separately. Equation (50) gives

$$e^Q = \frac{b^2}{X^2} \ln \frac{X^2}{C^2}. \tag{58}$$

Inserting these results in the metric (25), we have explicitly

$$g_{vX} = \frac{b^2 \ln^2 \frac{X}{C}}{X^2}, \tag{59}$$

$$g_{vv} = \frac{b^4 \left(2 - (a_1 + 2) \ln \frac{X}{C}\right) \ln^3 \frac{X}{C}}{a_2 X^2}. \tag{60}$$

In particular, these solutions involve logarithmic functions that diverge for vanishing $C$. For this reason we do not study them further in the present work.

### 4.2.3 Negative discriminant

Finally, for $\Delta < 0$, equation (49) gives

$$\tilde{X} = -\frac{a_1 + 2}{2a_2} - \frac{\sqrt{|\Delta|}}{2a_2} \tan\left[\frac{\sqrt{|\Delta|}}{2} \ln \frac{X}{C}\right]. \tag{61}$$

Integrating (50), one deduces

$$e^Q = \frac{b^2}{X^2} \cos^2\left[\frac{\sqrt{|\Delta|}}{2} \ln \frac{X}{C}\right]. \tag{62}$$

These solutions involve periodic functions in the dilaton field. It is unclear whether such solutions are physically relevant, so we shall not study them further in the present work.

## 5 AdS$_2$-to-dS$_2$ models

From now on, we focus on solutions with positive discriminant $\Delta > 0$. We will show that they have striking properties, including (i) a well-defined vacuum, (ii) an interpolation between an AdS$_2$ or flat region in the asymptotics and a dS$_2$ region in the center, and (iii) physically meaningful thermodynamical properties.

In section 5.1 we consider geometric, in particular asymptotic, aspects of these solutions, recast into Schwarzschild-like gauge. In section 5.2 we analyze the Ricci scalar, with particular focus on asymptotic values, values at the horizon and for vanishing dilaton. In section 5.3 we study the variational principle and holographically renormalize the action, assuming the boundary to be a dilaton iso-surface. In section 5.4 we exploit the previous results to derive the free energy and other thermodynamical quantities.

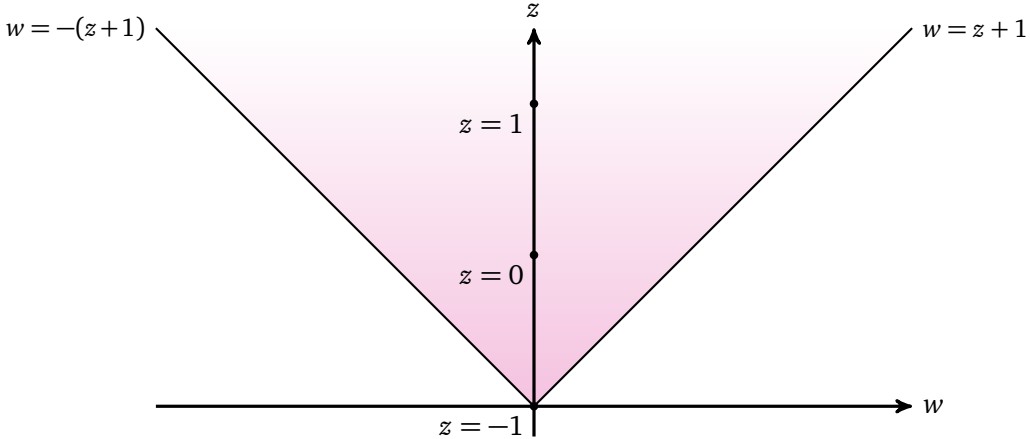

Figure 2: Solutions with horizon are in colored domain of $(w, z)$-parameter space

Before starting, it is convenient to recombine the parameters $\Delta, a_1, a_2$ as

$$z := \sqrt{\Delta} - 1, \qquad w := a_1 + 2, \qquad y^2 := \frac{a_2}{a_1 + 2 - \sqrt{\Delta}}. \tag{63}$$

Note that $y$ has dimension of length, and $z, w$ are dimensionless; as we shall demonstrate, $z$ is a Lifshitz-like anisotropy coefficient.

## 5.1  Asymptotic behavior, Lifshitz-like scaling and ground state

Let us start from the solution (55). By studying the zeros for $X$ of the equation $g_{vv} = 0$, one can show that there is a horizon at

$$X_H = C \left( \frac{z + 1 + w}{z + 1 - w} \right)^{\frac{1}{z+1}}, \tag{64}$$

provided

$$|w| < z + 1. \tag{65}$$

Furthermore, requiring Lorentz signature outside the horizon implies

$$y > 0. \tag{66}$$

Finally, we set

$$b = 1, \tag{67}$$

since we found that this parameter does not play any role in the analysis; it corresponds to a choice of time unit. The solutions are summarized in Fig. 2. The right diagonal in the figure lies outside our class of models and corresponds to the AdS–Schwarzschild–Tangherlini black branes discussed in appendix A.

Rewriting the metric (55) in a Schwarzschild-like gauge

$$\mathrm{d}s^2 = \Omega(X) \left( -\xi(X) \, \mathrm{d}t^2 + \frac{1}{\xi(X)} \, \mathrm{d}X^2 \right), \tag{68}$$

with

$$\Omega = \frac{|z|^{z+1}}{z} \left( \frac{X_H}{y} \right)^{2(z+1)} \left( \frac{z + 1 - w}{z + 1 + w} + \frac{X^{z+1}}{X_H^{z+1}} \right)^2 \left( \frac{X}{y} \right)^{-(z+3)}, \tag{69a}$$

$$\xi = \frac{|z|^{z+1}}{z} \left( \frac{X_H}{y} \right)^{z+1} \left( \frac{z + 1 - w}{z + 1 + w} + \frac{X^{z+1}}{X_H^{z+1}} \right) \left( 1 - \frac{X_H^{z+1}}{X^{z+1}} \right), \tag{69b}$$

is convenient for studying thermodynamical properties of the solution, which we shall do in section 5.4.

Expanding the solution (68) for large dilaton

$$g_{tt} = -z^{2z}\left(\frac{X}{y}\right)^{2z} - z^{2z}\left(3\frac{z+1-w}{z+1+w}-1\right)\left(\frac{X_H}{y}\right)^{z+1}\left(\frac{X}{y}\right)^{z-1} + \mathcal{O}(X^{-2}), \tag{70a}$$

$$g_{XX} = \left(\frac{y}{X}\right)^2 + \left(\frac{z+1-w}{z+1+w}+1\right)\left(\frac{X_H}{y}\right)^{z+1}\left(\frac{y}{X}\right)^{z+3} + \mathcal{O}\left(X^{-2(z+2)}\right), \tag{70b}$$

establishes the following properties of the metric: (i) it is field-independent at leading order, which allows to have a good notion of time asymptotically, no matter the specific value of the parameter $X_H$, (ii) the limit $z \to 0$ is regular and leads to an asymptotically flat metric, and (iii) for finite $z$, the metric $g_{\mu\nu}$ exhibits anisotropic scale invariance with Lifschitz-like exponent $z$. Indeed, the leading order terms in (70) are preserved by the anisotropic dilatation Killing vector

$$\xi_{\text{Lif}} = -zt\,\partial_t + X\,\partial_X\,. \tag{71}$$

The isotropic case is obtained for $z = 1$.

For a fixed value of the parameters $w$, $z$ and $y$, the vacuum of the theory is found by setting $X_H = 0$. The expressions (69) simplify to

$$\Omega = \frac{|z|^{z+1}}{z}\left(\frac{X}{y}\right)^{z-1}, \qquad\qquad \xi = \frac{|z|^{z+1}}{z}\left(\frac{X}{y}\right)^{z+1}, \tag{72}$$

and yield the ground state metric

$$ds_{\text{vac}}^2 = -\frac{z^{2z}}{y^{2z}}X^{2z}\,dt^2 + y^2\frac{dX^2}{X^2}\,. \tag{73}$$

This corresponds to Poincaré patch of AdS$_2$ spacetime with AdS radius $\ell = \frac{y}{z}$ when $z$ is finite and non-zero, and to Poincaré patch of Minkowski spacetime in the limit $z \to 0$. Note that global AdS$_2$ is not part of the solution space. The anisotropic Killing vector (71) generates an isometry of the ground state metric (73). In total, the ground state metric (73) has three Killing vectors,

$$L_{-1} = \partial_t\,, \qquad L_0 = -t\,\partial_t + \frac{X}{z}\,\partial_X\,, \qquad L_1 = t^2\,\partial_t - \frac{2tX}{z}\,\partial_X + \frac{y^{2z+2}}{z^{2z+2}}X^{-2z}\,\partial_t\,, \tag{74}$$

and hence is maximally symmetric. The ground state isometry algebra

$$[L_n, L_m] = (n-m)L_{n+m}\,, \qquad n, m \in \{-1, 0, 1\}\,, \tag{75}$$

is given by sl$(2, \mathbb{R})$ regardless of the value of $z \neq 0$ (and an İnönü–Wigner contraction thereof for $z \to 0$). As expected, for linear dilaton vacua (see e.g. [39]), the isometry algebra is broken to $u(1)$ by the dilaton, which is only annihilated by $L_{-1}$.

## 5.2 Ricci scalar

In two dimensions, spacetime curvature is completely encoded in the Ricci scalar. For the solutions (68) the Ricci scalar is given by

$$R = -\frac{2}{y^2}\left(\frac{X^{z+1}}{X_H^{z+1}} + \frac{z+1-w}{z+1+w}\right)^{-2}\left[z^2\left(\frac{X}{X_H}\right)^{2(z+1)} + \frac{(z+1)^2-2w}{w+z+1}\left(\frac{X}{X_H}\right)^{z+1} - \frac{z+1-w}{z+1+w}(2+z)^2\right]. \tag{76}$$

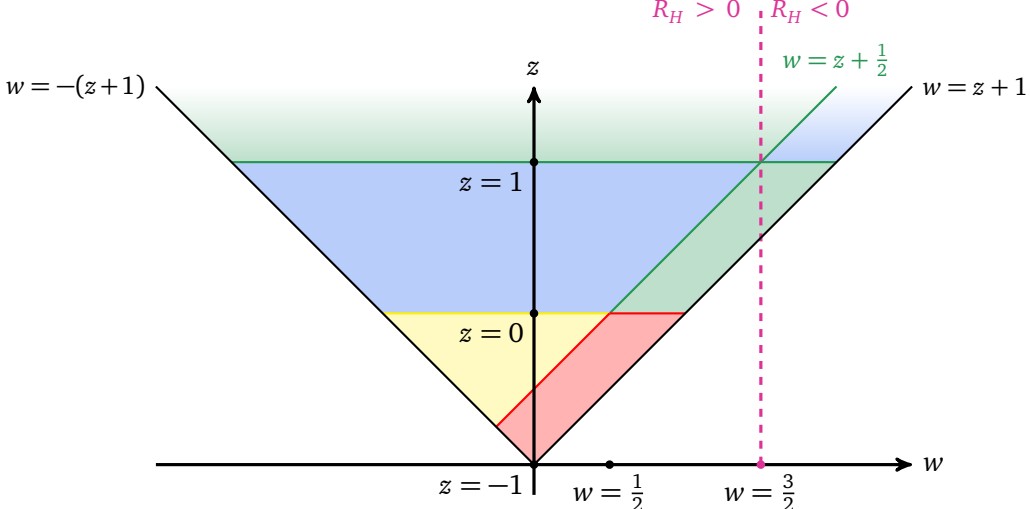

Figure 3: Four domains in $(w, z)$-parameter space with different qualitative Ricci behavior

In particular, at the origin and at the asymptotic boundary, the Ricci takes finite values

$$R_0 = \lim_{X \to 0} R = \frac{2(z+2)^2(z+1+w)}{y^2(z+1-w)}, \qquad R_\infty = \lim_{X \to \infty} R = -\frac{2z^2}{y^2} =: -\frac{2}{\ell^2}. \qquad (77)$$

Hence, the solutions are asymptotically AdS$_2$ when $\ell$ is finite (which happens for any positive $z$) and become asymptotically Minkowski in the limit $z \to 0$. The constraint (65) implies positive Ricci scalar at the origin $X \to 0$. Thus, generic solutions (with $z \neq 0$) interpolate between an asymptotically AdS$_2$ region and a dS$_2$ region. This is the main property of the Ricci scalar that we want to highlight. The Penrose diagram for black hole solutions with finite $X_H$ is depicted in Fig. 1.

Below we discuss further properties. The value of the Ricci scalar at the horizon (64),

$$R_H = -\left(w - \tfrac{3}{2}\right) \frac{(1+z+w)}{y^2}, \qquad (78)$$

is positive for $w < \frac{3}{2}$, negative for $w > \frac{3}{2}$, and vanishes for $w = \frac{3}{2}$. The Ricci scalar vanishes at one specific value of the dilaton,

$$X_{R=0}^{z+1} = \frac{X_H^{z+1}}{2z^2(w+z+1)} \Big[ 2w - (z+1)^2 \\ + (z+1)\Big( \sqrt{-4w^2(z(z+2)-1) - 4w + z(z+2)(4z(z+2)+1)+1} \Big) \Big]. \qquad (79)$$

For $X_H \to 0$ (and $X$ finite), the Ricci scalar (76) is constant and takes the same value,

$$\lim_{X_H \to 0} R = -\frac{2z^2}{y^2} = -\frac{2}{\ell^2}, \qquad (80)$$

as for AdS$_2$ metric (assuming finite $\ell$).

The qualitative behavior of the Ricci scalar as function of the parameters $z$ and $w$ is summarized in Fig. 3, with the shape of the Ricci scalar for each colored area depicted in Fig. 4.

We distinguish four domains:

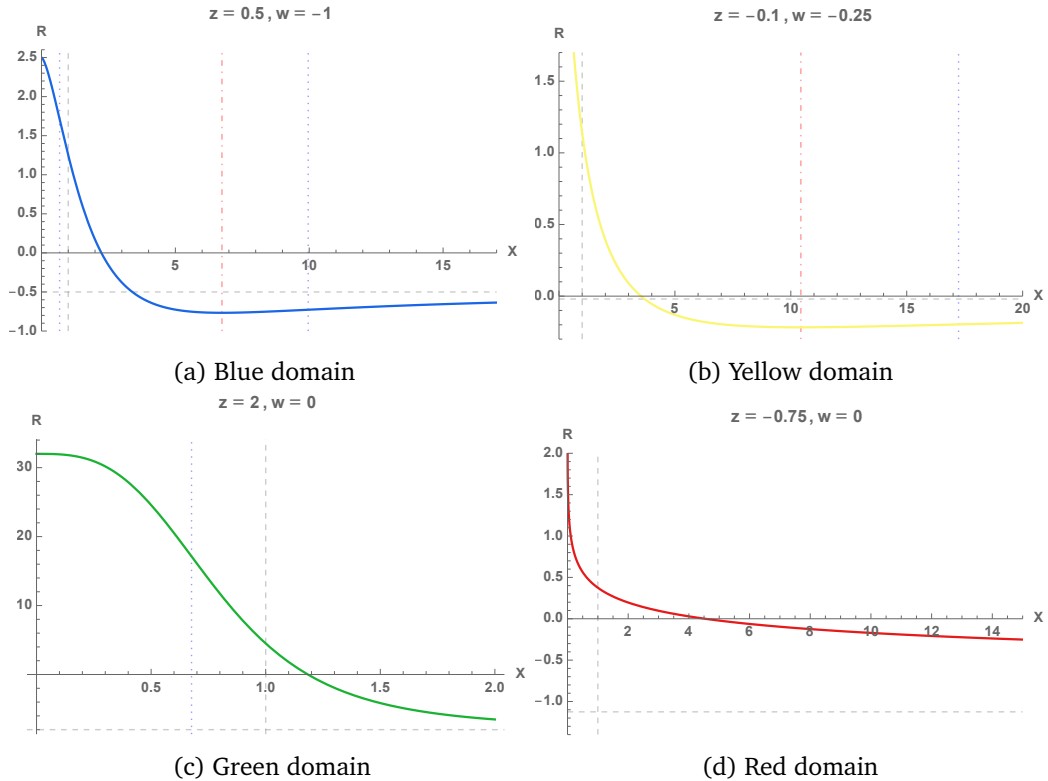

(a) Blue domain

(b) Yellow domain

(c) Green domain

(d) Red domain

Figure 4: Illustration of Ricci scalar behavior in the four domains ($X_H = 1, y = 1$). Grey vertical (horizontal) dashed lines denote horizons (asymptotic Ricci values). Red dot-dashed (blue dotted) lines show minimum (inflection point)

- Blue domain (middle left and upper right): this subsector lies in the domain $(z > 1, z + \frac{1}{2} < w < z + 1) \cup (0 < z < 1, -(z + 1) < w < z + \frac{1}{2})$. The Ricci scalar admits one minimum at

$$X_{\min}^{z+1} = X_H^{z+1} \frac{(z + 3)\left(2w^2 + w - 2z^2 - 5z - 3\right)}{(z - 1)(w + z + 1)(-2w + 2z + 1)}. \tag{81}$$

Furthermore, it has two inflection points $X_{\mathrm{infl}}^{\pm}$ such that $X_{\mathrm{infl}}^{-}, X_{R=0} < X_{\min} < X_{\mathrm{infl}}^{+}$.

- Yellow domain (lower left): this subsector lies in the domain $(-(z+1) < w < z + \frac{1}{2}, z \le 0)$. The Ricci scalar admits one minimum (81) and one inflection point $X_{\mathrm{infl}}^{+}$ such that $X_{R=0} < X_{\min} < X_{\mathrm{infl}}^{+}$.

- Green domain (upper left and middle right): this subsector lies in the domain $(-(z + 1) < w, w \le z + \frac{1}{2}, z \ge 1) \cup (0 < z \le 1, z + \frac{1}{2} \le w < z + 1)$. The Ricci has no minimum, but it admits an inflection point $X_{\mathrm{infl}}^{-}$ such that $X_{\mathrm{infl}}^{-} < X_{R=0}$.

- Red domain (lower right): this subsector lies in the domain $(-(z + 1) < w < z + 1, z + \frac{1}{2} \le w, z \le 0)$. The Ricci has neither minimum, nor inflection point. In particular, for $z = 0$, the Ricci vanishes only asymptotically, i.e., $X_{R=0}$ defined in (79) goes to infinity.

The Ricci curvature tends to a sign function when taking the limit $z \to \infty$ while keeping $w$ and $\ell$ finite

$$R\big|_{z \to \infty} = -\frac{2}{\ell^2} \operatorname{sign}(X - X_H). \tag{82}$$

The transition between positive and negative curvature regions matches with the position of the horizon, i.e., $X_{R=0} \to X_H$ (see Fig. 5). In other words, there is a sharp transition from dS$_2$

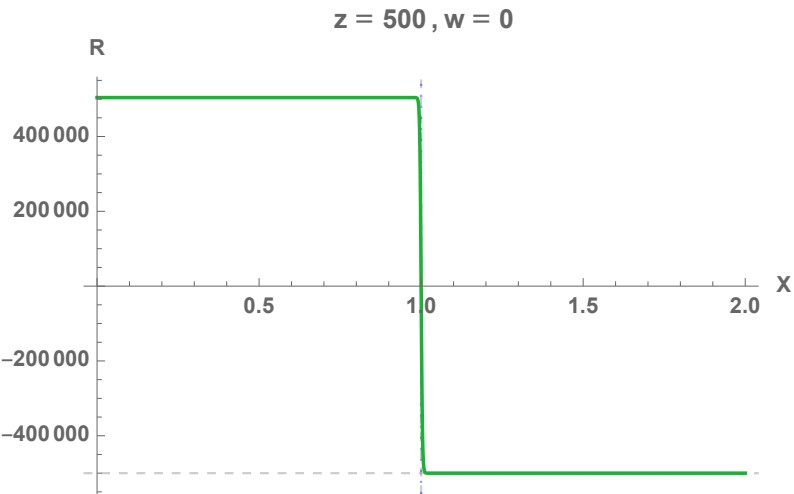

Figure 5: Ricci scalar approaches sign function with jump localized at horizon

in the interior to asymptotically AdS$_2$ outside the black hole for large anisotropy coefficient, $z \gg 1$. The steep gradient near the horizon is reminiscent of the large $D$-expansion of general relativity [21, 22, 50, 51], suggesting that $1/z$ could be fruitfully used as perturbation parameter for black hole perturbations, backreactions, etc. More precisely, the large $D$ limit can be understood as a double limit $w - 1 \to z \to \infty$, according to the example (121) in appendix A.

Finally, we exploit our results to discuss potential cosmological applications, e.g. along the lines of [52]. To be more specific, we pose the question whether there exists a range of parameters for which the region around the horizon has almost constant positive curvature, so that it can be approximated as a dS$_2$ horizon. To this end we define the dimensionless quantity

$$\sigma := \left| \frac{\mathrm{d} \ln R}{\mathrm{d} \ln X} \right|. \tag{83}$$

If both the inequalities $\sigma|_{X_H} = (2w^2 + w - 2(z(z+2)+2))/(2w-3) \ll 1$ and $w < \frac{3}{2}$ are satisfied, the horizon is within a region of approximately constant positive curvature, since the (positive) Ricci scalar varies slowly as a function of the dilaton. However, these inequalities [together with the convexity condition (65)] are never satisfied simultaneously, since for $w < \frac{3}{2}$ the quantity $\sigma|_{X_H}$ is bounded from below, $\sigma|_{X_H} \geq \frac{1}{2}$. Hence, a positive Ricci scalar always changes considerably in the region around the horizon. (By contrast, we can have an approximately AdS$_2$ horizon for $w > \frac{3}{2}$.)

The quantity (83) can also be computed in the limits $X \to 0$ and $X \to \infty$. We find $\sigma \to 0$ in both cases, which means that the center of the spacetime and infinity are constant curvature regions. This concurs with our previous conclusion that all solutions interpolate between a dS$_2$ region in the center and an asymptotic AdS$_2$ region for $z > 0$.

## 5.3 Holographic renormalization

In this section, we perform the holographic renormalization of the action along the lines of [53]. Specifically, we work with metrics of the form (68) that are well adapted for this study. Having in mind thermodynamical applications, we work with Euclidean time coordinate $\tau = it$ and identify periodically $\tau \sim \tau + 2\pi\beta$ with $\beta = T^{-1}$. We assume the boundary to be a dilaton iso-surface, i.e., $\delta X|_{\partial M} = o(X)$.

The first term in the action,

$$I = -\frac{\kappa}{4\pi} \int_M d^2x \sqrt{g} \left[ RX - 2\mathcal{V}(X, -(\partial X)^2) \right] - \frac{\kappa}{2\pi} \int_{\partial M} dx \sqrt{\gamma} XK, \qquad (84)$$

is the bulk action for generalized dilaton gravity (29). The second term is the Gibbons–Hawking–York-like boundary term, suitable for a Dirichlet boundary value problem; $K = \nabla_\mu n^\mu$ is the extrinsic curvature and $n^\mu$ the normal to the boundary $\partial M$. The variation of the action (84) yields on-shell

$$\delta I \approx \frac{\kappa}{2\pi} \int_{\partial M} dx \sqrt{\gamma} \left( \pi^{ab} \delta \gamma_{ab} + \pi_X \delta X \right), \qquad (85)$$

where

$$\pi^{ab} = -\frac{1}{2} \gamma^{ab} n^\mu \nabla_\mu X, \qquad \pi_X = -2(\partial_\nabla \mathcal{V}) n^\mu \nabla_\mu X - K. \qquad (86)$$

We need to add a boundary counter term $I_{ct}$ to the action (84) so that the total action,

$$\Gamma = I + I_{ct}, \qquad (87)$$

has a well-defined variational principle, $\delta\Gamma \approx 0$, for all solutions (68). This will additionally ensure finiteness of $\Gamma$ when evaluated on solutions (68).

The bare on-shell action, denoted by $I_{reg}$, is evaluated at a certain cutoff $X = X_R$

$$I_{reg} \approx \frac{\kappa\beta}{2\pi} \frac{|z|^{z+1}}{z} \left[ -\left(\frac{X_R}{y}\right)^{z+1} + \frac{2(z+1)^2 - 4w}{2(w+z+1)} \left(\frac{X_H}{y}\right)^{z+1} \right.$$
$$\left. + \frac{z+1-w}{z+1+w} \left(\frac{X_H}{y}\right)^{2(z+1)} \left(\frac{y}{X_R}\right)^{z+1} \right]. \qquad (88)$$

The counter-term that ensures a well-defined variational principle,

$$I_{ct} = \frac{\kappa}{2\pi y} \int_{\partial M} d\tau \sqrt{\gamma} X, \qquad (89)$$

is essentially the same as for the JT model. Once the counter-term (89) is added to the action, the cutoff can be removed, $X_R \to \infty$, yielding a finite on-shell action (87)

$$\left. \Gamma \right|_{EOM} = -\frac{\kappa\beta}{4\pi} \left( 1 + \frac{z+1-w}{z+1+w} \right) |z|^z z \left(\frac{X_H}{y}\right)^{z+1}. \qquad (90)$$

By construction, the action (87) with the counter-term (89) has a well-defined variational principle, $\delta\Gamma \approx 0$.

## 5.4 Thermodynamics

Using the results of the previous section we investigate now the thermodynamics of the solutions (68). We start with the temperature, continue with the mass and finish with entropy, free energy and specific heat.

The Hawking–Unruh temperature of the solutions (68) is determined from the Euclidean metric

$$ds^2 = f(r) d\tau^2 + \frac{1}{h(r)} dr^2, \qquad f(r) = \Omega(r)\xi(r), \quad h(r) = \Omega^{-1}(r)\xi(r). \qquad (91)$$

The horizon (64) is a (simple) zero of $f(r)$ and $h(r)$. Expanding around $r = X_H$, $f(r) = f'(X_H)(r - X_H)$, $h(r) = h'(X_H)(r - X_H)$ and performing the coordinate transformation $\rho = 2\sqrt{(r - X_H)/(h'(X_H))}$, $\theta = \frac{\tau}{2}\sqrt{f'(X_H)h'(X_H)}$ the near horizon metric $d\rho^2 + \rho^2 d\theta^2$

is regular at $\rho = 0$ provided $\theta \sim \theta + 2\pi$. This implies the periodicity of Euclidean time $\beta = 4\pi / \sqrt{f'(X_H)h'(X_H)}$ the inverse of which is the Hawking–Unruh temperature

$$T = \frac{(z+1)^2 |z|^z}{2\pi(z+1+w)y^{z+1}} X_H^z. \tag{92}$$

Temperature (92) has length dimension $-z - 1$, and the same is true for the mass defined below.

Evaluating the general expression for the charges (37) on the solutions (68) for the Killing vector $\xi = \partial_\tau$ associated with Euclidean time translations obtains the mass

$$M = \frac{\kappa(z+1)|z|^z}{2\pi(z+1+w)y^{z+1}} X_H^{z+1} \stackrel{(64)}{=} \frac{\kappa(z+1)|z|^z}{2\pi(z+1-w)y^{z+1}} C^{z+1}. \tag{93}$$

The vacuum (73) has vanishing mass, $M = 0$. Furthermore, one can check that the first law is satisfied,

$$\delta M = T \delta S, \tag{94}$$

with the Wald entropy [54]

$$S = \kappa X_H. \tag{95}$$

The scaling of entropy with mass

$$S \sim M^{\frac{1}{z+1}} \tag{96}$$

is compatible with the expected one for theories with anisotropic scale invariance, see e.g. [55–57].

The free energy deduced from the results above,

$$F = M - TS = -\frac{\kappa}{2\pi} \frac{z(z+1)|z|^z}{(z+1+w)y^{z+1}} X_H^{z+1}, \tag{97}$$

is consistent with the on-shell value of the holographically renormalized action (90), i.e.,

$$F = \beta^{-1} \Gamma\big|_{\text{EOM}}. \tag{98}$$

Specific heat for positive $z$ is given by[6]

$$\mathcal{C} = T \frac{\partial S}{\partial T} = \frac{\kappa}{z^2} \left( \frac{2\pi(w+z+1)y^{z+1}}{(z+1)^2} \right)^{1/z} T^{1/z}. \tag{99}$$

It is always positive, showing perturbative thermodynamical stability of our black hole solutions. In the limit $z \to 0^+$ temperature (92) is state-independent and the inverse specific heat vanishes, which are features shared with the CGHS model [58].[7]

Since the free energy (97) of all black hole states is negative, while the vacuum free energy is zero, the black holes are also stable non-perturbatively, i.e., there is no Hawking–Page like phase transition.

---

[6]For negative $z$ specific heat has the same magnitude as in (99) but a negative sign. Thus, for negative $z$ all black hole solutions are perturbatively thermodynamically unstable, a feature shared with the Schwarzschild black hole.

[7]The same is true for the $\widehat{\text{CGHS}}$ model [59], but not for its twisted version [60].

# 6 Boundary conditions and asymptotic symmetries

In this section we investigate some boundary conditions and the associated asymptotic symmetries for the scale invariant dilaton gravity models. Specifically, we focus on models with the potential (44) of the quadratic form (47). We found that the asymptotic analysis depends drastically on the choice of Lifschitz scaling $z$.

In the present work we focus on the isotropic case, $z = 1$ (this implies $\ell = y$). The main goal of this section is to show that this case exhibits the following features: (i) the asymptotic symmetry algebra generates $\mathrm{Diff}(S^1) \ltimes C_\infty(S^1)$ asymptotic symmetries, (ii) the boundary conditions involve fluctuating dilaton at leading order, and (iii) the charges are finite and can be made integrable after suitable field-dependent redefinitions of the parameters, leading to the Heisenberg algebra.

## 6.1 Solution space

Our boundary conditions for the metric and the dilaton,

$$g_{tt} = -\frac{r^2}{\ell^2} + g_{tt}^{(0)}(t) + \mathcal{O}\left(r^{-2}\right), \qquad g_{rr} = \frac{\ell^2}{r^2} + g_{rr}^{(4)}(t)\frac{\ell^4}{r^4} + \mathcal{O}\left(r^{-6}\right), \qquad (100a)$$

$$g_{tr} = \frac{2}{r}\left(\ell^2\Phi'(t) - \Psi(t)\right) + \mathcal{O}\left(r^{-3}\right), \qquad X = e^{\Phi(t)}\, r, \qquad (100b)$$

are motivated by the falloff behavior of the solutions in Schwarzschild-like gauge (70).[8] The state-dependent functions $\Phi(t)$ and $\Psi(t)$ can be freely chosen, while the remaining functions are determined through on-shell conditions from them. Namely, the EOM (31) together with the quadratic potential (44), (47) imply

$$g_{tt}^{(0)} = \ell^2\left((w+1)\Phi'^2 - \Phi''\right) + \frac{4(w-1)}{\ell^2}\Psi^2 + (w-1)g_{rr}^{(4)} + 2\left((1-2w)\Phi'\Psi + \Psi'\right), \quad (101a)$$

$$g_{rr}^{(4)} = -\ell^2\Phi'^2 - \frac{4}{\ell^2}\Psi^2 + \frac{4\pi}{\kappa}M\,e^{-2\Phi} + 4\Phi'\Psi, \qquad (101b)$$

$$M' = 0. \qquad (101c)$$

Here, $M$ is a constant of motion coinciding with the mass (93), related to the Casimir $C$ by $M = \frac{\kappa}{\pi\ell^2}\frac{1}{(2-w)}C^2$. In particular, one recovers the solution (70) by setting $\Phi = 0 = \Psi$.

Since some length dimensions are unusual we mention them explicitly: $t$ has dimension 2; $\ell$ has dimension 1; $r, X, \Phi, \Psi, C, \kappa$ are dimensionless; $M$ has dimension $-2$.

## 6.2 Asymptotic Killing vectors

The boundary conditions (100) are preserved under diffeomorphisms generated by

$$\zeta = \left(F(t) + G(t)\frac{1}{r^2} + \mathcal{O}\left(r^{-4}\right)\right)\partial_t + \left(-F'(t)r - \Phi'(t)G(t)\frac{1}{r} + \mathcal{O}\left(r^{-3}\right)\right)\partial_r. \qquad (102)$$

At leading order, these asymptotic Killing vectors satisfy the Lie-bracket relations

$$\left[\zeta(F_1, G_1), \zeta(F_2, G_2)\right] = \zeta\left(F_1 F_2' - F_2 F_1', (F_1 G_2)' - (F_2 G_1)'\right). \qquad (103)$$

In Euclidean signature, the boundary is $S^1$ and this algebra corresponds to $\mathrm{Diff}(S^1) \ltimes C_\infty(S^1)$. The state-dependent functions transform as

$$\delta_\zeta\Phi = -F' + F\Phi', \qquad \delta_\zeta\Psi = -\frac{G}{\ell^2} - \frac{\ell^2}{2}F'' + (\Psi F)', \qquad (104)$$

---

[8]The quantity $\tilde{X}$ is determined in terms of the dilaton $X$ and the Casimir $C$ by (49).

while the Casimir is invariant, $\delta_\zeta C = 0$. The latter statement follows trivially from the Poisson $\sigma$-model formulation, since in Casimir–Darboux coordinates the Poisson tensor has a row and column of zeros corresponding to the Casimir direction, so that the analogue of the left transformation equation (6) implies Casimir invariance under all gauge transformations (including improper ones).

## 6.3 Charges and Cardy formula

Inserting the solutions described in section 6.1 and the asymptotic Killing vectors derived in section 6.2 into the general expression (37) and evaluating it at $r \to \infty$ yields the variation of the charges[9]

$$\frac{2\pi}{\kappa} k^{tr}_{\text{BB},\zeta} = \delta e^\Phi \, \delta_\zeta \Psi - \delta_\zeta e^\Phi \, \delta \Psi + F \, e^{-\Phi} \frac{2\pi}{\kappa} \, \delta M \, . \tag{105}$$

This expression is finite in the radial expansion parameter $r$ but not integrable in field space. A procedure to render these charges integrable is described below. The charges consists of two qualitatively different pieces: (i) a boundary contribution coming from the term $F \, e^{-\Phi} \frac{2\pi}{\kappa} \, \delta M$ and (ii) a corner contribution coming from the part $\delta e^\Phi \, \delta_\zeta \Psi - \delta_\zeta e^\Phi \, \delta \Psi$.

The charges (105) were computed in the second order formulation, using the Barnich–Brandt expression (37). One can show that the $E$-term (38) relating the Barnich–Brandt and the Iyer–Wald procedures does not contribute to the charge expression for $r \to \infty$, which implies $k^{tr}_{\text{BB},\zeta} = k^{tr}_\zeta$ at the spacetime boundary.

If one specifies the analysis to the stationary solutions (70), the charges (105) reduce to

$$k^{tr}_{\text{BB},\zeta}\big|_{\text{stationary}} = F \, \delta M \, , \tag{106}$$

which reproduces the mass (93) when taking $F = 1$. Furthermore, the entropy (95) can be rewritten suggestively as

$$S = 2\pi\ell \sqrt{\frac{c \, M}{6}} \, , \qquad c = \frac{3\kappa}{2\pi} (w + 2) \, , \tag{107}$$

which is (a chiral version of) the Cardy formula. Since the mass $M$ is just a constant, it is fair to ask whether there is a Virasoro tower $L$ that reduces to $M/\ell^2$ for zero mode solutions and that transforms with an infinitesimal Schwarzian

$$\delta_\zeta L = F \, L' + 2F' \, L + \frac{c}{12} F''' \, , \tag{108}$$

with the central charge $c$ given in (107). The answer is affirmative and can be deduced by analogy to section 5 in [61], yielding

$$L = \frac{M}{\ell^2} e^{-2\Phi} - \frac{\kappa}{16\pi} (w + 2) \big( \Phi'^2 + 2\Phi'' \big) \, . \tag{109}$$

## 6.4 Heisenberg charges and algebra

Finally, we resolve the non-integrability of the charges (105). As there are no local physical degrees of freedom in generalized dilaton gravity models, according to [62, 63] there exists an integrable slicing of the state space. Technically, this means that there must exist a field-dependent redefinition of the symmetry generators, such that the resulting charges are integrable. We show now by explicit construction that this is indeed the case.

---

[9]See appendix B for the first order analysis.

Considering

$$\lambda_q = e^{2\Phi}\left(\frac{G}{\ell^2} + \frac{\ell^2}{2}F'' + F'\Psi - F\Psi' - 2F\Psi\Phi'\right), \qquad \lambda_M = e^{-\Phi}F, \qquad (110)$$

and defining $\lambda_p = \lambda'_M$, $q = e^{-\Phi}$ and $p = e^{2\Phi}\Psi$, the charge variation reduces to

$$\frac{2\pi}{\kappa}k^{tr}_{\text{\tiny BB},\zeta} = \lambda_q\,\delta q + \lambda_p\,\delta p + \lambda_M\frac{2\pi}{\kappa}\,\delta M\,. \qquad (111)$$

Since we define $\lambda_{p,q,M}$ to be state-independent, we can integrate (111) in field space to obtain the boundary charges

$$\mathcal{Q}[\lambda_q, \lambda_p, \lambda_M] = \frac{\kappa}{2\pi}\left(\lambda_q\,q + \lambda_p\,p\right) + \lambda_M\,M \qquad (112)$$

that obey the Heisenberg algebra

$$\delta_\zeta q = \lambda_p\,, \qquad \delta_\zeta p = -\lambda_q\,. \qquad (113)$$

Of course, the Casimir, and hence the mass, remain invariant for this (or any other) slicing, $\delta_\zeta C = \delta_\zeta M = 0$. In appendix B we show that the same results can be obtained in the first order formulation.

This particular slicing of the state space where the charge algebra reduces to the Heisenberg algebra is sometimes called fundamental slicing [62, 63]. It arose first in the context of near horizon boundary conditions and soft hair excitations in three-dimensional Einstein gravity [64], and it has also been shown to appear in the asymptotic boundary analysis of two-dimensional spacetimes [46]. Another similarity to higher-dimensional gravity (see [65, 66] and refs. therein) is that the entropy is linear in the Casimir and blind to the (soft hair) excitations generated by $q$ and $p$. This comparison suggests to reinterpret the Casimir $C$ as the near horizon Hamiltonian and to implement the near horizon soft hair program in (generalized) dilaton gravity in two dimensions.

# 7 Outlook

Generalized dilaton gravity in two dimensions (2) provides numerous research avenues. As an outlook, we list a few of them below, without claiming to be exhaustive. We start with more specific issues related to the dilaton scale invariant models on which we focused.

- **SYK-like correspondences.** Given the developments in the past half decade (see [67] and the reviews [68–70]) it seems natural to attempt finding an SYK-like model [71, 72] dual to the $z = 1$ model studied in section 6 and check e.g. whether or not it saturates the chaos bound [73, 74] and/or can be interpreted as matrix model [75].

- **Schwarzian type boundary actions.** An important link between gravity and field theory is the boundary action describing (from a gravity perspective) the dynamics of edge modes. Since the JT analysis of the Schwarzian boundary action [76] generalizes e.g. to higher spin theories [77] and to flat space dilaton gravity [59], it is plausible that some (or even all) of the models presented in this work also lead to boundary actions that have a physical interpretation in terms of edge modes and a mathematical interpretation as some group action associated with certain coadjoint orbits.

- **Holographic renormalization for fluctuating dilaton.** In section 5.3 we assumed dilaton iso-surfaces as boundary to obtain a holographically renormalized action suitable for deriving the free energy. However, in SYK-like contexts this assumption is dropped and the dilaton is allowed to fluctuate at the boundary, which leads to a kinetic boundary term [78,79]. For the class of models introduced in section 4 it is plausible that similar terms are needed. It could be instructive to construct them.

- **Boundary conditions for $z \neq 1$.** The boundary conditions discussed in section 6 are valid for the isotropic case, $z = 1$. It might be rewarding to generalize this discussion to the anisotropic case, $z \neq 1$, to check, among other things, whether or not a Lifshitz-type Cardy-like formula for the entropy emerges and if again the Heisenberg algebra pops up. Such an analysis may also help to understand the somewhat mysterious origin of the Lifshitz anisotropy in these models.

- **Large $z$ limit and horizon localization.** The sign-function behavior of the Ricci scalar (82) at large $z$ means that all non-trivial aspects of geometry localize near the horizon, reminiscent of what happens in the large $D$-limit of general relativity [21, 22]. When adding matter to the system, it could be useful to treat $1/z$ as small parameter and simplify the technical discussion of black hole formation, stability and evaporation. Moreover, there might be a simplified "membrane theory" that localizes around the horizon and that captures the essential features of black holes. This route may provide models of comparable simplicity and universality as the CGHS model [58] or the Almheiri–Polchinski model [80, 81]. In order to proceed along these lines in a meaningful way one would have to add matter to the system, see below.

- **Other singular limits.** Recently, various limits — including non- and ultra-relativistic ones — of the JT model and generalizations thereof were derived [82, 83]. The same limits can be applied to generic models (2), as indicated already in [82]. It should be possible (and might reveal interesting new aspects) to be exhaustive in classifying all meaningful limits of generalized dilaton gravity; part of the challenge is to be precise what "meaningful" means in this context.

- **Applications of dS-AdS.** In our work we have merely observed that solutions in a certain parameter range interpolate between dS in the IR and AdS in the UV. However, we have not attempted to exploit this fact for studying physical aspects of dS holography or toy model cosmology. While we are not confident that the specific models studied in section 4 will be useful for this purpose, it seems likely that within the generic class of models (2) there will be useful ones that could allow a refinement of the discussion initiated in [52].

- **Generic dilaton scale invariant models.** Scale invariant models of type (1) were studied recently from a holographic perspective in [84]. It could be of interest to generalize their discussion (and ours) to generic dilaton scale invariant models (44).

We conclude with more general remarks that go beyond the dilaton scale invariant models.

- **Bottom-up model building.** The model space provided by (2) is not only infinitely larger than the one provided by (1), it also yields qualitatively new features that may be propitious for (toy) model building. To give one specific example, for all models (1) it is true that the Weyl factor [denoted by $Q(X)$] appearing in the metric depends only on the dilaton and some rather irrelevant integration constant, but not on the mass of the state. By contrast, for generic models (2) the function $Q(X)$ also depends on the Casimir (and hence the mass of the state); see for example the explicit construction in section

4.2. This qualitatively new feature may allow for (families of) hitherto unknown models that highlight certain aspects of gravity, holography and black hole physics not captured by the more restrictive class (1).

- **Top-down model constructions.** While we have no concrete proposal, it is conceivable that the higher derivative terms allowed by the action (2) may appear in some top-down constructions, say, from string theory. Indeed, the developments of black holes in string theory [18–20, 85] were intimately related with some of the developments in two-dimensional dilaton gravity, see e.g. [86].

- **Charting the model space.** All models (2) are deformations of each other in a technical sense, but this does not imply that any two given models are physically close to each other. For instance, geometric properties such as the asymptotic behavior, number and types of Killing horizons, presence or absence of singularities, as well as the precise form of the boundary action and the associated asymptotic symmetries depend on the choice of the function $\mathcal{V}$. It could be valuable to either find a measure for the proximity of a model or to group all models into suitable subclasses, such that within a given subclass all models are reasonably similar to each other. The attribute "similarity" might be defined in terms of intrinsically two-dimensional entities, such as the asymptotic behavior or the boundary action, or in terms of external structures, such as possible lifts to a specific higher dimension or reformulations in terms of matrix models. See, for instance, the discussions in [8, 87, 88].

- **Matter, backreactions and black hole evaporation.** As alluded to in some of the previous items, an exciting prospect is to add some matter action to the geometric one (2) to address issues like black hole formation, stability and evaporation. Some of the developments for usual dilaton gravity (1) are summarized in the review [7]. Basically, one could go through this review line-by-line and attempt to generalize the results to generic dilaton gravity with matter. Given the qualitative changes mentioned above this route could be more than just work-therapy and lead to novel insights into evaporating black holes. Of course, also more recent developments could be generalizable, such as the island proposal [89, 90] (see also [91] and refs. therein).

- **Entropy universality.** The results of section 6.4 suggest that the Casimir $C$ has a physical interpretation as near horizon Hamiltonian. The appearance of the Heisenberg algebra and the fact that entropy is linear in this putative near horizon Hamiltonian indicates a universal feature of generalized dilaton gravity that rhymes well with higher-dimensional constructions of soft Heisenberg hair [64, 92] and points towards a universality of entropy, $S \sim C$, in all gravity theories (including higher spins [93], higher derivatives [94], higher form degrees [95], and higher dimensions [66]). It should be possible to test entropy universality, as well as the related conjecture [96] that entropy is slicing-independent, for generalized dilaton gravity (2).

- **Origin of the corner term.** We derived the corner term needed to connect the first and second order symplectic potentials. It could be interesting to study whether it has a geometric interpretation in terms of boundary quantities along the lines of [97–99].

In summary, exciting new possibilities are provided by generalized dilaton gravity in two dimensions (2).

## Acknowledgements

We are grateful to Hamid Afshar, Dio Anninos, Glenn Barnich, Florian Ecker, Gaston Giribet, Bob McNees, Stefan Prohazka, Jakob Salzer, Shahin Sheikh-Jabbari and Dima Vassilevich for discussions.

**Funding information** This work was supported by the Austrian Science Fund (FWF), projects M 2665, P 30822, P 32581 and P 33789. Research at Perimeter Institute is supported in part by the Government of Canada through the Department of Innovation, Science and Economic Development Canada and by the Province of Ontario through the Ministry of Colleges and Universities.

## A Geometries for power-counting renormalizable models

For pedagogical reasons, in this appendix we apply the general algorithm of section 2.3.1 to the power-counting renormalizable models (1), thereby recovering well-known results, see e.g. [7, 36, 37].

The potential $\mathcal{V}$ is given by

$$\mathcal{V} = -X^+X^-U(X) + V(X).\tag{114}$$

The algorithm works exactly as explained in the main text, so the only new information provided here are explicit integrations. In particular, (20) simplifies to

$$\mathrm{d}(X^+X^-) + (X^+X^-U(X) - V(X))\,\mathrm{d}X = 0,\tag{115}$$

which can be integrated explicitly upon introducing the integrating factor

$$Q(X) := \int^X U(y)\,\mathrm{d}y,\tag{116}$$

yielding

$$\mathrm{d}C = 0, \qquad C := e^{Q(X)}X^+X^- + w(X),\tag{117}$$

where

$$w(X) := -\int^X e^{Q(y)}V(y)\,\mathrm{d}y.\tag{118}$$

The quantity $C$ is the Casimir, which on-shell is conserved in time and space (117).

The quantity $Q(X)$ defined in (116) coincides with the corresponding quantity in (24). Therefore, the explicit form of the metric (25),

$$\mathrm{d}s^2 = 2\,\mathrm{d}v\,\mathrm{d}r - 2e^{Q(X(r))}(w(X) - C)\,\mathrm{d}v^2,\tag{119}$$

depends on the value of the Casimir (117) and on the two functions (116) and (118), which are uniquely determined from the dilaton potentials $U(X)$ and $V(X)$, up to irrelevant integration constants. The radial coordinate $r$ is related to the dilaton by

$$\mathrm{d}r = e^{Q(X)}\,\mathrm{d}X,\tag{120}$$

which can be integrated to express the dilaton as function of the radius, $X = X(r)$. Evidently, all solutions (119), (120) have at least one Killing vector, the coordinate vector $\partial_v$, which proves explicitly the generalized Birkhoff theorem for power-counting renormalizable models.

Note that in this special class of models the function $Q(X)$ is independent from the Casimir. As mentioned in the main text, this ceases to be true for generic models (2).

An even more concrete class of examples is provided by AdS–Schwarzschild–Tangherlini black branes in $D$-dimensional Einstein gravity, dimensionally reduced to two spacetime dimensions, assuming $D > 3$. In this case, the potentials are given by

$$U(X) = -\frac{D-3}{(D-2)X}, \qquad V = -\frac{(D-1)(D-2)}{2}\frac{X}{\ell^2}, \qquad (121)$$

recovering the JT model for $D \to 3$ [100] and the Witten black hole for $D \to \infty$ [51]. These models lie on the right diagonal in Figs. 2 and 3. Inserting the potentials (121) into the expressions in this appendix yields the dilaton

$$X = \left(\frac{r}{D-2}\right)^{D-2}, \qquad (122)$$

and the metric

$$ds^2 = 2\,dv\,dr - \left(\frac{r^2}{\ell^2} - \frac{2M}{r^{D-3}}\right)dv^2, \qquad (123)$$

where the mass parameter $M$ is linearly related to the Casimir,

$$M = (D-2)^{D-3}C. \qquad (124)$$

## B Comparison with charges in first order formalism

In this appendix, we compare the results for charges between first and second order formulations, using as basis the boundary analysis of section 6 in the second order formulation.

To proceed with the first order formulation, we choose the dyad

$$\begin{pmatrix} e_r^+ & e_r^- \\ e_t^+ & e_t^- \end{pmatrix} = \begin{pmatrix} \frac{g_{rr}}{2} & 1 \\ \frac{1}{2}\left(g_{rt} + \sqrt{|g|}\right) & \frac{1}{g_{rr}}\left(g_{rt} + \sqrt{|g|}\right) \end{pmatrix}, \qquad (125)$$

where the components of the metric $g_{\mu\nu}$ are given in (100). On-shell, we have

$$e_r^+ = \frac{1}{2r^2} + \left(\frac{\kappa}{2\pi}Me^{-2\Phi} - \frac{\left(\Phi' - 2\Psi\right)^2}{2}\right)\frac{1}{r^4} + \mathcal{O}\left(r^{-6}\right), \qquad (126a)$$

$$e_t^+ = -\frac{1}{2} + \frac{\Phi' - \Psi}{r} + \mathcal{O}(r^{-2}), \qquad (126b)$$

$$e_r^- = 1, \qquad (126c)$$

$$e_t^- = r^2 + 2r(\Phi' - \Psi) + \mathcal{O}(r^0), \qquad (126d)$$

where we set $\ell = 1$ throughout this appendix. The gauge symmetries in the first order formulation are the combination of Lorentz transformations (10) and diffeomorphisms (11), viz., $\delta_{\xi,\lambda_\omega}e^a = \mathcal{L}_\xi e^a - \lambda_\omega \epsilon^a{}_b e^b$. They preserve the solution space (126) provided the diffeomorphism parameters $\xi$ are given by (102) and the Lorentz parameter satisfies

$$\lambda_\omega = F + \frac{2G}{r} + \mathcal{O}\left(\frac{1}{r^2}\right). \qquad (127)$$

The Lorentz parameter does not bring additional symmetries into the analysis and is determined from the diffeomorphism generators (102).

Computing the corner term (42) for the solution space (126) in the limit $r \to \infty$ yields

$$Y^{tr}[\phi, \delta\phi] = \frac{\kappa}{2\pi} e^{\Phi} \delta\left(\Phi' - \Psi\right).$$

(128)

Interestingly, this term does not vanish, implying that the charge computation in first and second order formalism does not produce the same charges. In particular, this contribution cancels the first two terms of the second line of (105) corresponding to the Heisenberg part of the charge algebra. Explicitly, the charges in the first order formulation are obtained by evaluating the general expression (34) for our solution space and evaluating it at the boundary,

$$\frac{2\pi}{\kappa} k^{tr}_{1,\zeta,\lambda_\omega} = \delta\left[-2e^{\Phi}\right]G + \delta\left[e^{\Phi}(2\Psi - \Phi')\right]F' + \delta\left[\frac{\kappa}{2\pi}M\right]e^{-\Phi}F$$
$$+ 2e^{\Phi}(\delta\Phi\,\Psi' - \delta\Psi\,\Phi') + e^{\Phi}\left(\frac{1}{2}\delta(\Phi')^2 - \Phi''\delta\Phi\right)F,$$

(129)

where the index 1 stands for first order. Again, as in the second order formalism, the charges are finite but not integrable. They are related to the charges obtained in (105) via the corner term (128) as

$$k^{tr}_{\text{BB},\zeta}[\phi, \delta\phi] = k^{tr}_{1,\zeta,\lambda_\omega}[\phi, \delta\phi] - (\delta_{\zeta,\lambda_\omega} Y^{tr}[\phi, \delta\phi] - \delta Y^{tr}[\phi, \delta_{\zeta,\lambda_\omega}\phi]).$$

(130)

By analogy to the main text, we render the charge expression (129) integrable by performing field-dependent redefinitions of the symmetry parameters,

$$\lambda_{1,q} = e^{2\Phi}\left(2G - F'\Phi' + 2\Psi\left(F' - 2F\Phi'\right) + F\left(2(\Phi')^2 + \Phi'' - 2\Psi'\right)\right),$$

(131)

$$\lambda_M = e^{-\Phi}F,$$

(132)

and $\lambda_p = \lambda'_M$. Using similar definitions as in section 6.4, $q = e^{-\Phi}$ and $p_1 = e^{2\Phi}\left(2\Psi - \Phi'\right)$, the charges reduce to

$$\frac{2\pi}{\kappa} k^{tr}_{1,\zeta,\lambda_\omega} = \lambda_{1,q}\,\delta q + \lambda_p\,\delta p_1 + \frac{2\pi}{\kappa}\lambda_M\,\delta M,$$

(133)

and satisfy the algebra

$$\delta_\zeta q = \lambda_p, \qquad\qquad \delta_\zeta p_1 = -\lambda_{1,q},$$

(134)

which, again, corresponds to the Heisenberg algebra.

For this particular example, we have shown that the first and second order formulations lead to the same number of parameters ($F$ and $G$) for the residual diffeomorphisms. From general considerations on auxiliary fields, it is expected that the number of parameters is the same when dealing with exact symmetries (see e.g. [101]). However, when treating asymptotic symmetries, this is not guaranteed *a priori* (one could have additional parameters appearing in the local Lorentz transformations) and one has to verify it explicitly, which is what we did above. See [99] for a similar analysis in higher dimensions.

Another interesting feature is that although the respective solution spaces and residual gauge symmetries are the same in first and second order formulations, their symplectic structures differ. This difference is controlled by the corner term (42) appearing at the level of the Lagrangians. Despite of this discrepancy, after judiciously redefining the symmetry parameters, the charge algebras are identical in the two formulations and yield the Heisenberg algebra. This could have been anticipated, since the charges form a representation of the asymptotic symmetry algebras, which are the same in the two formulations.

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
