# Peer review of "Generalized dilaton gravity in 2d"

_SciPost Physics, doi:SciPost Phys. 12, 032 (2022)_

## Round 2 · Referee Report · Anonymous · 2021-10-22

Strengths
1. Very clearly written
2. Finds an interesting class of new models with intriguing properties
3. Raises many interesting directions to follow up on
Weaknesses
1. Could benefit from more details on some aspects of the models/solutions, see report
Report
This paper studies two dimensional dilaton-gravity theories. A large new class of such models is introduced in the paper. These models are obtained from Poisson sigma models, a class of generalized gauge theories, by imposing that local Lorentz invariance is part of the gauge group. This presents the gravitational model in a first order formulation and it is shown that in the second order formulation these models correspond to having an arbitrary dilaton potential that can depend on the magnitude of the gradient of the dilaton besides the dilaton itself. Allowing such potentials results in models that are not power-counting renormalizable, but it is argued that this is not a problem since the theories possess no local degrees of freedom. The authors argue that this is the most general class of theories with this field and gauge content, based on a similar result about Poisson sigma models by Izawa. It is shown that these theories always admit linear dilaton solutions, and the phase space and symplectic form are constructed explicitly.
The rest of the paper focuses on a three parameter sub-family of these new models, with one parameter introducing the minimal new feature, i.e. gradient of the dilaton to the fourth power. The linear dilaton solutions are explicitly written down for this sub-family. There are three distinct cases depending on the value of a discriminant formed from the three parameters of the model. A well defined asymptotic region and vacuum exists for positive discriminant and the rest of the paper focuses on this case.
The authors proceed to analyze the linear dilaton solutions and they find that, depending on the values of the parameters of the model, they can correspond to asymptotically $AdS_2$ or flat black hole solutions. Moreover, near the origin (defined where the dilaton is zero), the solution looks like de Sitter, and the geometry can be positively curved at the horizon. The behavior of the Ricci scalar in different regions of the spacetime is analyzed in detail as a function of the parameters of the model. The vacuum of these theories is Poincare $AdS_2$. For generic values of the parameters, the black hole solutions are found to be Lifshitz-like, in the sense that there is a non-trivial scaling exponent between the mass and the temperature. In the limit of large “anisotropy”, the solutions limit to a locally $AdS_2$ spacetime in the exterior, glued to a locally $dS_2$ spacetime in the interior along the horizon.
The authors then proceed to discuss the Euclidean solution and examine its thermodynamic properties, by calculating the renormalized on-shell action, the Hawking temperature, and the Wald entropy of the horizon, these fit together consistently as expected. The solutions are found to be thermodynamically stable.
Finally, the boundary conditions and asymptotic symmetries are laid out for the isotropic models, and the corresponding asymptotic charges are constructed.
Two dimensional dilaton-gravity theories are very useful arenas to tackle non-perturbative questions about gravity. JT gravity and some of its generalizations have been subject to much recent interest in this regard, providing new insights into the black hole information problem, spacetime emergence in AdS/CFT, and raising new puzzles about quantum gravity. This paper provides a large extension of this class of models, working out many of the classical aspects of the interesting solutions, paving a way to a quantum study. Moreover, extremely intriguing features are found, such as the de Sitter interiors, and the Lifschitz scaling in the absence of boundary anisotropy, all deserving further study. I therefore heartily recommend this excellent paper for publication.
I have a couple of small questions that the authors could optionally consider addressing:
-What are $a_0$ and $a_1$ in (12)? They are left undefined but clash with the parameters of the three parameter model in sec. 5.
-Is it obvious that the Izawa proof generalizes in the presence of the local Lorentz invariance constraint on $P_{IJ}$?
-Is JT gravity reachable as a limiting case of the positive discriminant class? The fact that $R>0$ at the origin always suggests that it’s not, since in JT gravity $R$ is constant negative everywhere. Similar comments are made in the paper about the zero discriminant case, but this point about JT gravity maybe worth emphasizing. Could there be other such “islands” in model space whose solutions cannot be deformed into each other?
-How does the Kruskal extension/Penrose diagram of these solutions look like? Is it a two boundary strip as in the case of JT gravity, with alternating $X\rightarrow\infty$ and $X\rightarrow-\infty$ boundaries, but now with de Sitter-like interior regions?
-The discussion of Lifschitz scaling around (71) seems maybe a bit misleading, after all, this anisotropy near the boundary maybe removed by a change of coordinates (the same change that puts the vacuum $AdS_2$ in standard Poincare coordinates). Of course, comparing (92) and (93) or looking at (96) shows clearly the Lifschitz-like physical features.
-For negative $z$ in empty $AdS_2$ (73), the origin and the asymptotic boundary are swapped, which may affect the Euclidean analysis in the BH case (among other things). Does this affect the conclusion of negative heath capacity for this branch?
-Where was the dilaton scale invariance (45) important for the paper?
-In the asymptotic symmetry group, the factor of $C_\infty(S^1)$ presumably refers to the exponentiated version of $G $not being required to be bijective. How can one see that there are no such global constraints on the finite versions of $G$-transformations?
-Before (105): yields the charges->yields the variation of the charges?

---

## Round 2 · Referee Report · Anonymous · 2021-10-30

Strengths
1. A large new class of models of dilaton gravity is derived, which might be interesting.
2. The work is structurally tentalizing and raises immediate follow-up questions.
Weaknesses
1. The calculations in section 2.3, around which much of the article centers, are densely written.
2. It is not yet clear if the new models are actually physically interesting, or whether this will end up being more a classification exercice.
Report
This work explores the space of physically sensible models of dilaton gravity. The idea is to start with the known gauge theory (or first order) formulation of JT gravity, then generalize the underlying gauge theoretic structure from a BF theory to something known as a Poisson sigma model. They then impose that the Poisson sigma is actually describing gravity by demanding local Lorenz invariance. This makes sense, and by rewriting things in the second order fomulation (metric and dilaton) they find a general class of dilaton gravity actions corresponding with those sigma models - more general than the theories considered earlier.
They then solve the classical equations of motions, but without using any boudnary conditions, which I find peculiar, but one could choose to do this. Section 2.3 where they solve the EOM is in my opinion too densely written.
They then focus on a certain subset of new models, one could say the "most obvious" generalizations beyond the class of theories studied before, which can essentially be captured by one deformation parameter in (48). The classical solutions can have horizons, and can have a dS region behind the horizon whilst being asymptotically AdS. This last observation is surprising, and the main argument in favor of considering these deformations as interesting.
It was not clear to me what to make of this observation, but it sounds potentially interesting and could potentially turn out to be useful.
The work raises interesting follow up questions. From a structural point of view, two obvious questions are (1) whether one can go beyond classical solutions and use the Poisson sigma model formulation to compute path integrals exactly (as you can do for JT and many other dilaton gravity models), and (2) if with the appropriate boundary conditions the model as a holographic dual as quantum mechanics on (a constrained version of) the target space of the sigma model. From a physical perspective the obvious question is what to make of this dS region.
Because the work raises these interesting follow-up questions, I recommend it for publication.
I do have some recomendations / minor complaints that I think would increase the quality of the paper
1. The meaning of the notation $d\pm\omega$ in (19b) and (19d) is unclear to me.
2. Section 2.3 is quite densely written, perhaps a small appendix would be usefull.
3. There are a lot of symbols for the solutions being used, some of which are redundant (for example on could eliminate $y$ in function of $\ell$), this makes it a bit thougher for the reader to understand what is physically happening.
4. It is unclear why the authors focus on $z=1$ in section 6 as "preferred" value. To me it lookes as if the regime $z=\infty$ is most insteresting, given formula (82).
5. It is mentionned at some point that C is related to the mass, it would be useful to refer to the relevant equation there, it took me a while to find it.

---

## Round 3 · Author Response

We thank both referees for their suggestions and questions, most of which we have addressed in our amended version of the manuscript. Below we reply to each of them:
Report 1:
1.) What are a0 and a1 in (12)? They are left undefined but clash with the parameters of the three parameter model in sec. 5.
We have adapted the notation to avoid potential confusion by putting tildes on top of these quantities. Moreover, we have expanded the statement after (12) to be a bit more explicit about a0 and a1.
2.) Is it obvious that the Izawa proof generalizes in the presence of the local Lorentz invariance constraint on PIJ ?
Yes. Izawa's proof is for general PSMs, while we are additionally imposing a restriction of local Lorentz invariance. Imposing this restriction and deforming commutes, since after any deformation we have to make sure that the constraint is met again. We did not change anything in the paper regarding this issue, since we are already fairly explicit about solving the non-linear Jacobi identities and the implications of local Lorentz invariance.
3.) Is JT gravity reachable as a limiting case of the positive discriminant class? The fact that R>0 at the origin always suggests that it’s not, since in JT gravity R is constant negative everywhere. Similar comments are made in the paper about the zero discriminant case, but this point about JT gravity maybe worth emphasizing. Could there be other such “islands” in model space whose solutions cannot be deformed into each other?
While JT gravity is within the positive discriminant class (a_2=a_1=0, a_2\neq 0), we have assumed in and after (48) that a_2\neq 0. (and we stated this assumption explicitly). Within the AdS_2-to-dS_2 models studied from section 5 onwards, JT is a singular limit. In particular, the parameter y in (63) vanishes for JT and thus the Ricci scalar (76) is ill-defined. Nevertheless, it is possible to take a JT limit directly for the Ricci scalar (76) if it is expressed in terms of the AdS radius. Since this issue resonated with others raised by the second referee we added statements concerning JT and, more generally, Schwarzschild-Tangherlini black branes (including the Witten black hole) in the paragraph below (67) and the new appendix A. From this discussion, it becomes clear that JT is not an "island" in our model space, but rather lies on its boundary (just outside of it).
4.) How does the Kruskal extension/Penrose diagram of these solutions look like? Is it a two boundary strip as in the case of JT gravity, with alternating X→∞ and X→−∞ boundaries, but now with de Sitter-like interior regions?
The answer is provided by Fig. 1, which is the Penrose diagram for the black hole solutions discussed in sections 5-6. To highlight this better we added a sentence at the end of the first paragraph of section 5.2.
5.) The discussion of Lifschitz scaling around (71) seems maybe a bit misleading, after all, this anisotropy near the boundary maybe removed by a change of coordinates (the same change that puts the vacuum AdS2 in standard Poincare coordinates). Of course, comparing (92) and (93) or looking at (96) shows clearly the Lifschitz-like physical features.
While we agree with the referee's sentiment that the very notion of Lifhsitz in a 2d gravity context may be misleading (after all, in the dual QFT there are not sufficiently many directions with respect to which there could be an anisotropy), we believe that we have stated the case fairly: the main evidence for Lifshitz-like behavior comes from the scaling symmetries after (71) and especially from the thermodynamical behavior (96); we concede in the discussion already that more needs to be done "to understand the somewhat mysterious origin of the Lifshitz anisotropy in these models". Therefore, we did not change anything concerning this point, but we certainly agree that understanding this Lifshitz scaling will require new insights.
6.) For negative z in empty AdS2 (73), the origin and the asymptotic boundary are swapped, which may affect the Euclidean analysis in the BH case (among other things). Does this affect the conclusion of negative heath capacity for this branch?
Yes, this is one way of understanding why the specific heat becomes negative. Since we address negative z already in footnote 6 (and we are unsure how relevant this case is physically) we did not add anything on top of that.
7.) Where was the dilaton scale invariance (45) important for the paper?
Foremost, it was a selection criterion to reduce the model space and study a concrete class of models with this additional (generalized) symmetry. Technically, dilaton scale invariance implied that state-dependence encoded in the Casimir C always appeared multiplicatively with the dilaton, i.e., in the form of functions that depended on X/C. It is a fair question whether there is more to be said on dilaton scale invariance. Currently, we do not have anything intelligent to add, so we did not change anything in the paper regarding this point.
8.) In the asymptotic symmetry group, the factor of C∞(S1) presumably refers to the exponentiated version of Gnot being required to be bijective. How can one see that there are no such global constraints on the finite versions of G-transformations?
While the finite transformations and the group structure associated with the symmetry algebra are of interest, we have focused on the infinitesimal version. Note that C_\infty(S^1) represents the algebra of functions on the circle. The complete finite analysis is beyond the scope of the work. Indeed, we are interested in the organization of the phase space which is governed by the asymptotic symmetry algebra discussed in the paper. Currently, we have nothing additional to say on this issue.
9.) Before (105): yields the charges->yields the variation of the charges?
Yes, thanks! Corrected in the updated version.
Report 2:
- The meaning of the notation d±ω in (19b) and (19d) is unclear to me.
We are unsure what the confusion is about, so to be very explicit: d denotes the exterior derivative, appearing for the first time in Eq. (4) \pm is the standard symbol for either + or - (so Eqs. (19b) and (19d) are two Eqs. each, with upper and lower signs) \omega denotes the dualized spin connection, appearing for the first time a few lines above Eq. (8)
Since all the symbols are pretty standard and have appeared several equations before (19), we did not change anything here.
- Section 2.3 is quite densely written, perhaps a small appendix would be useful.
We have added Appendix A, performing explicitly the two remaining integrals for the class of power-counting renormalizable models (1). Moreover, we included as a special family of examples the dimensionally reduced Schwarzschild-Tangherlini black branes in D spacetime dimensions, including the limits of JT (D \to 3) and Witten black hole (D \to \infty). We hope that the referees (and other readers) find this new Appendix useful.
- There are a lot of symbols for the solutions being used, some of which are redundant (for example on could eliminate y in function of ℓ), this makes it a bit thougher for the reader to understand what is physically happening.
We sympathize with this suggestion and had corresponding discussions when writing the paper. However, note that y cannot be eliminated for ℓ, since their relationship is singular for z=0, which lies within the considered model space. Therefore, despite some apparent redundancy, we stick with our conventions.
- It is unclear why the authors focus on z=1 in section 6 as "preferred" value. To me it lookes as if the regime z=∞ is most insteresting, given formula (82).
We did not claim that z=1 is preferred, though perhaps it would even be correct to make such a claim. [After all, only for z=1 isotropy of the scale invariance is restored and the generalized Cardy-formula reduces to the traditional (chiral half of the) Cardy-formula.] The point of section 6 was not to be encyclopedic, but merely to present one specific example to highlight relevant features of boundary conditions, charges, their non-integrability, and how to overcome it by changing the slicing. The lessons drawn from this section apply more generically, but details of the calculations need to be adapted on a case-by-case basis. Concerning z\to\infty, that limit probably is most interesting when coupling our model to matter in order to study, e.g., black hole formation, evaporation, and/or backreaction, since without matter all models are exactly soluble anyhow. Most likely, perturbation theory in 1/z only becomes a powerful tool when exact solutions cease to be available. Studying the coupling of our models to matter is something we left to future work.
- It is mentionned at some point that C is related to the mass, it would be useful to refer to the relevant equation there, it took me a while to find it.
Thanks, point taken! We have added a second equality in Eq. (93) explicitly relating the mass M to the constant C.

---

## Round 3 · List of Changes

*) Eq. (12) and 1-3 lines after that Eq.: put tildes on top of a_1 and a_0
*) after Eq. (12): replaced "... the most general solutions for a_0, a_1, and deduced \tilde L = ..." by "... the most general solutions for the bottom of the descent ladder, the ghost-number 2 term \tilde a_0, as well as the higher steps in the ladder, \tilde a_0 and \tilde L = ..."
*) in the paragraph below (67) added "The right diagonal in the figure lies outside our class of models and corresponds to the AdS–Schwarzschild–Tangherlini black branes discussed in appendix A."
*) at the end of the first paragraph of section 5.2 added "The Penrose diagram for black hole solutions with finite $X_H$ is depicted in Fig.~1."
*) on page 18 added: "More precisely, the large D limit can be understood as a double limit w − 1 → z → ∞, according to the example (121) in appendix A."
*) in Eq. (93): second equality added to express M in terms of C.
*) before Eq. (105): replaced "yields the charges" by "yields the variation of the charges"
*) added Appendix A

---

## Editorial Decision

published